# Role of the pre-initiation complex in Mediator recruitment and dynamics

**Elisabeth R Knoll[1], Z Iris Zhu[2], Debasish Sarkar[3], David Landsman[2], Randall H Morse[1,3]***

[1]Department of Biomedical Sciences, School of Public Health, University at Albany, Albany, United States; [2]Computational Biology Branch, National Center for Biotechnology Information, National Library of Medicine, Bethesda, United States; [3]Wadsworth Center, New York State Department of Health, Albany, United States

**Abstract** The Mediator complex stimulates the cooperative assembly of a pre-initiation complex (PIC) and recruitment of RNA Polymerase II (Pol II) for gene activation. The core Mediator complex is organized into head, middle, and tail modules, and in budding yeast (*Saccharomyces cerevisiae*), Mediator recruitment has generally been ascribed to sequence-specific activators engaging the tail module triad of Med2-Med3-Med15 at upstream activating sequences (UASs). We show that yeast lacking Med2-Med3-Med15 are viable and that Mediator and PolII are recruited to promoters genome-wide in these cells, albeit at reduced levels. To test whether Mediator might alternatively be recruited via interactions with the PIC, we examined Mediator association genome-wide after depleting PIC components. We found that depletion of Taf1, Rpb3, and TBP profoundly affected Mediator association at active gene promoters, with TBP being critical for transit of Mediator from UAS to promoter, while Pol II and Taf1 stabilize Mediator association at proximal promoters.
DOI: https://doi.org/10.7554/eLife.39633.001

**\*For correspondence:**
randall.morse@health.ny.gov

**Competing interests:** The authors declare that no competing interests exist.

## Introduction

The Mediator complex plays a central, highly conserved role in eukaryotic transcription by stimulating the cooperative assembly of a pre-initiation complex (PIC) and recruitment of RNA Polymerase II (Pol II) for gene activation (*Allen and Taatjes, 2015*; *Jeronimo and Robert, 2017*; *Kornberg, 2005*; *Malik and Roeder, 2010*; *Soutourina, 2018*). In budding yeast (*Saccharomyces cerevisiae*), Mediator is composed of 25 subunits divided into four domains based on structural and functional criteria: the tail, middle, head, and kinase modules (*Guglielmi et al., 2004*; *Plaschka et al., 2016*; *van de Peppel et al., 2005*). Mediator recruitment has generally been ascribed to sequence-specific activators engaging the tail module triad of Med2-Med3-Med15 (*Lee et al., 1999*; *Myers et al., 1999*; *Zhang et al., 2004*; *Ansari and Morse, 2013*); consistent with this model, the tail module can under some circumstances be recruited independently of the remainder of Mediator (*Ansari et al., 2009*; *He et al., 2008*; *Paul et al., 2015*; *Petrenko et al., 2016*; *Zhang et al., 2004*; *Anandhakumar et al., 2016*). Normally, however, the tail is recruited as a complex with the head, middle, and kinase modules at upstream activating sequences (UASs) (*Jeronimo and Robert, 2014*; *Wong et al., 2014*; *Grünberg et al., 2016*; *Kuras et al., 2003*). Once Mediator is recruited, the kinase module dissociates from the complex and Mediator bridges the enhancer to connect with Pol II and general transcription factors (GTFs) at the proximal promoter of active genes (*Jeronimo et al., 2016*; *Petrenko et al., 2016*). Mediator association with the PIC at proximal promoters is quickly disrupted following phosphorylation of the Pol II C-terminal domain (CTD) by Kin28, marking the end of initiation (*Jeronimo and Robert, 2014*; *Wong et al., 2014*).

This orchestrated recruitment of Mediator is broadly required for transcription, evidenced by studies that showed a loss of Mediator function results in greatly decreased mRNA levels and

concomitant diminished association of Pol II with essentially all Pol II transcribed genes (*Ansari et al., 2009*; *Holstege et al., 1998*; *Jeronimo and Robert, 2014*; *Paul et al., 2015*; *Plaschka et al., 2015*; *Thompson and Young, 1995*). In spite of the purported role of the tail module triad in Mediator recruitment by transcriptional activators, *med3Δ med15Δ* yeast are viable, show defects in mRNA levels for only 5–10% of active genes during normal growth, and exhibit continued Mediator and Pol II occupancy by ChIP-seq (*Ansari et al., 2012a*; *Jeronimo et al., 2016*; *Paul et al., 2015*; *Petrenko et al., 2016*). Additional studies report that the tail module is not required for Mediator's interaction with the PIC (*Jeronimo et al., 2016*; *Petrenko et al., 2016*; *Plaschka et al., 2015*). That transcription is unaffected at many genes in *med3Δ med15Δ* yeast supports the notion that Mediator is recruited through an unknown mechanism that is independent of the tail module, possibly through components of the PIC which are known to engage Mediator subunits at proximal promoters (*Ansari and Morse, 2013*; *Esnault et al., 2008*; *Soutourina et al., 2011*; *Eychenne et al., 2016*; *Larivière et al., 2006*). Indirect recruitment of Mediator via interactions with the general transcription machinery has also been proposed on the basis of in vitro experiments (*Johnson and Carey, 2003*). However, because Med2 is still present in *med3Δ med15Δ* mutants, the possibility that recruitment occurs via this remaining tail module triad subunit cannot formally be excluded.

Several studies show that a loss of Mediator leads to a decrease in PIC components and diminished Pol II presence at active genes, highlighting the unilateral importance of Mediator in stimulating the assembly of a PIC (*Eyboulet et al., 2015*; *He et al., 2008*; *Holstege et al., 1998*; *Paul et al., 2015*; *Thompson and Young, 1995*; *Kuras and Struhl, 1999*; *Li et al., 1999*). Yet, how the PIC itself influences Mediator recruitment has received little attention (*Eychenne et al., 2016*; *Robinson et al., 2016*). The close spatial relationship of Mediator with the PIC at proximal promoters has been shown in single particle cryo-electron microscopy and cross-linking mass spectrometry studies of yeast in which Mediator is bound with the PIC (*Nozawa et al., 2017*; *Plaschka et al., 2015*; *Robinson et al., 2016*; *Tsai et al., 2017*). The TFIID-TBP complex and Pol II in particular occupy proximal promoters and have the potential to engage Mediator directly. In vivo, however, Mediator's interaction with the PIC at proximal promoters is likely brief in light of recent evidence demonstrating that Mediator association at proximal promoters is transient, making interactions with the PIC difficult to assess in living cells (*Fan and Struhl, 2009*; *Jeronimo and Robert, 2014*; *Wong et al., 2014*).

To investigate PIC-dependent Mediator recruitment in vivo we have performed ChIP-seq against Mediator subunits in *Saccharomyces cerevisiae* after employing the anchor away method of nuclear depletion to export individual components of the PIC to the cytoplasm (*Haruki et al., 2008*). We confirm Mediator stabilization at proximal promoters in the *kin28*-AA strain. In the same background, we show that in *med2Δ med3Δ med15Δ* yeast, which completely lack the tail module triad, association of Mediator and Pol II is greatly reduced but nonetheless still detected at the majority of targets. We also investigate Mediator association by ChIP-seq after depletion of PIC components TBP, Rpb3, and Taf1, both alone and together with Kin28, to assess the contribution of the PIC to Mediator occupancy. Taken together, our findings provide new insights into mechanisms of Mediator recruitment during gene activation in yeast.

## Results

### The tail module triad is non-essential but is important for recruitment of Mediator and Pol II to all promoters

As outlined in the Introduction, recruitment of Mediator by a mechanism independent of the tail module triad has been proposed on the basis of experiments employing *med3Δ med15Δ* yeast mutants. However, the possibility that residual tail module function supported by Med2 is sufficient to allow Mediator and Pol II recruitment has not been formally excluded. Therefore, to assess more rigorously the role of the tail module triad in Mediator function, we sought to determine the effect of deletion of all three tail module triad subunits on yeast viability and Mediator recruitment.

Segregants from diploid yeast heterozygous for *MED2*, *MED3*, and *MED15* (complete ORF deletions) included viable *med2Δ med3Δ med15Δ* mutants, which showed modest growth defects similar to those seen for *med3Δ med15Δ* yeast (*Figure 1A*). RNA-seq showed that two independent

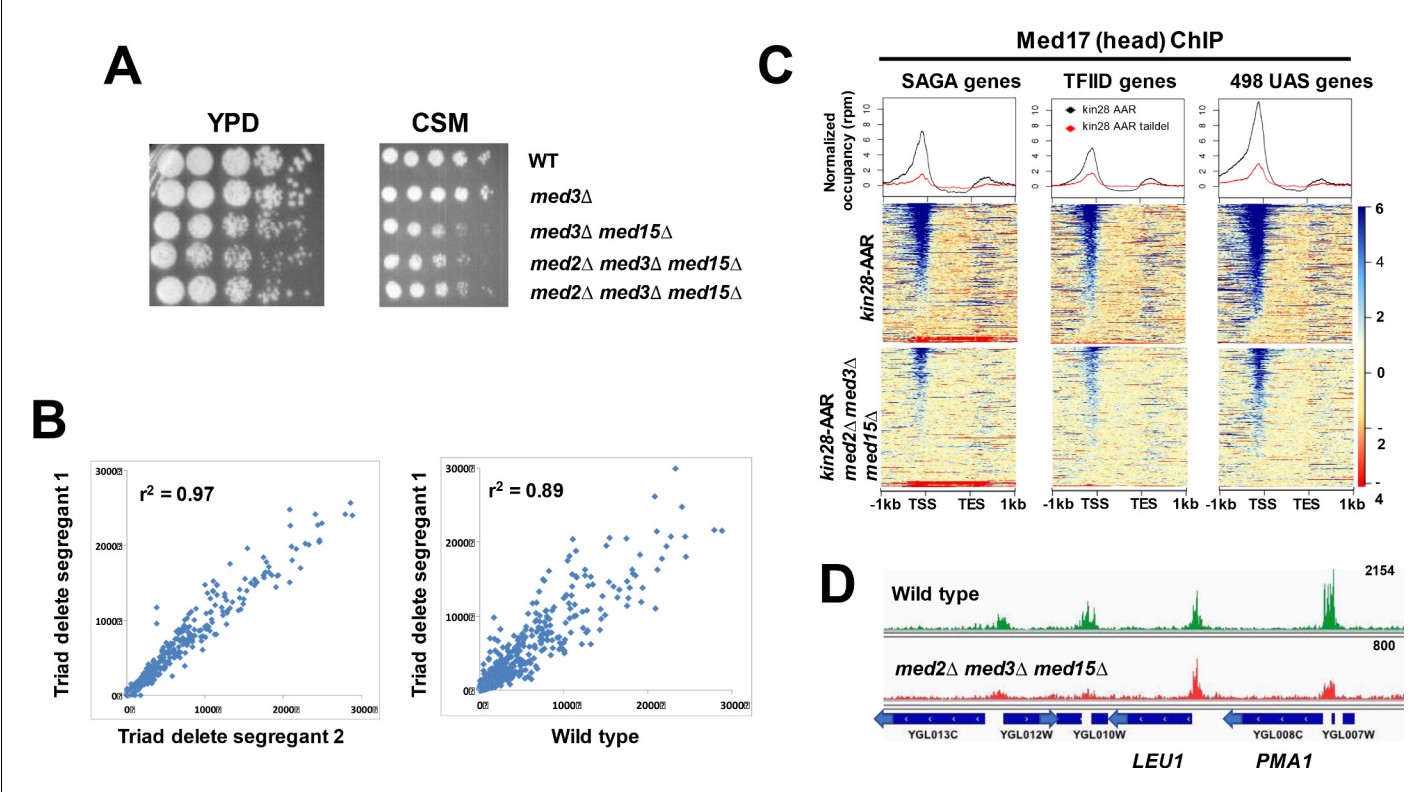

**Figure 1.** Mediator recruitment in *med2Δ med3Δ med15Δ* yeast. (**A**) Spot dilutions (five-fold from left to right) of the indicated strains were made on YPD or CSM plates and allowed to grow at 30 °C for 3 days. The two *med2Δ med3Δ med15Δ* strains are independent segregants. (**B**) Comparison of transcript levels (reads per million) by RNA-seq for two independent *med2Δ med3Δ med15Δ* segregants (left panel), and between segregant one and the wild type parent strain (right panel). Only data for mRNA genes are plotted, after normalization to total mRNA. (**C**) Normalized Med17 occupancy in *kin28-AA* and *med2Δ med3Δ med15Δ kin28-AA* yeast after 1 hr treatment with rapamycin (designated 'AAR'). Reads were mapped to all SAGA-dominated genes, all TFIID-dominated genes, and to the 498 genes exhibiting detectable Mediator ChIP signal at UAS regions in wild type yeast (*Jeronimo et al., 2016*). Genes were normalized for length and aligned by transcription start site (TSS) and transcription end site (TES), and are sorted according to average signal intensity. A compressed color scale was used for heat maps to better allow visualization of signal. (**D**) Browser scan showing normalized Med17 occupancy in wild type and *med2Δ med3Δ med15Δ* yeast over a short region of Chromosome VII; the upper end of the occupancy scale (reads) is indicated on the upper right for each graph. Note that the peak associated with *LEU1* (TFIID-dominated) is essentially unchanged in *med2Δ med3Δ med15Δ* relative to wild type yeast, while the two peaks at the left (associated with TFIID-dominated transcripts) and that associated with *PMA1* (SAGA-dominated) are all reduced in *med2Δ med3Δ med15Δ* yeast.

DOI: https://doi.org/10.7554/eLife.39633.002

The following figure supplement is available for figure 1:

**Figure supplement 1.** ChIP signal metagene and peak analysis in *kin28-AA* yeast after rapamycin treatment.

DOI: https://doi.org/10.7554/eLife.39633.003

segregants exhibited genome-wide expression patterns more similar to each other than to wild type yeast (*Figure 1B*) (Fisher's z-score = 37, $p < 10^{-10}$ for the null hypothesis that the correlations are not significantly different). Down-regulated genes were enriched for SAGA-dominated promoters (75 of 164 genes down-regulated by >2X in both segregants; $p < 10^{-34}$, hypergeometric test), while up-regulated genes showed marginal enrichment for SAGA-dominated promoters (35 of 442 genes, p=0.04), consistent with previous findings (*Ansari et al., 2012a*). RNA-seq also confirmed the triple deletion phenotype, with no reads present for sequences from the *MED2*, *MED3*, or *MED15* ORF.

We next examined the effect of loss of the tail module triad on Mediator recruitment. Mediator association with UAS elements is seen in ChIP experiments at only a fraction of active yeast genes, and its association with active promoters appears to be transient, making it difficult to assess Mediator association with gene regulatory regions in wild type yeast (*Fan et al., 2006*; *Fan and Struhl, 2009*; *Jeronimo and Robert, 2014*; *Paul et al., 2015*; *Wong et al., 2014*). Mediator ChIP signal at promoters is greatly enhanced under conditions in which phosphorylation of the Pol II CTD by the

Kin28 subunit of TFIIH is prevented, apparently by inhibiting escape of Pol II from the promoter proximal region (*Jeronimo and Robert, 2014*; *Wong et al., 2014*). We therefore constructed a Kin28 anchor away (*kin28-AA*) strain harboring the *med2Δ med3Δ med15Δ* mutations to assess Mediator occupancy genome-wide. Treatment of this strain with rapamycin tethers Kin28 to the Rpl13A ribosomal subunit via FKBP12 and FRB tags on the respective proteins, and leads to Kin28 eviction from the nucleus during ribosomal protein processing (*Haruki et al., 2008*). The strain also includes the *tor1-1* mutation, which prevents the normal stress response provoked by rapamycin (*Haruki et al., 2008*).

We used ChIP-seq to examine occupancy by the head module subunit Med17 (Srb4) (*Bourbon et al., 2004*) at SAGA-dependent and TFIID-dependent promoters (*Huisinga and Pugh, 2004*). Consistent with previous work, treatment of *kin28-AA* yeast harboring intact Mediator allowed detection of myc-tagged Med17 at the majority of promoters (*Figure 1C–D*) (*Jeronimo and Robert, 2014*; *Wong et al., 2014*), while parallel experiments using an untagged strain or an input sample gave rise to very low signal (*Figure 1—figure supplement 1A*). Mediator peaks observed after depletion of Kin28 corresponded well with those identified in a previous study (*Figure 1—figure supplement 1B*) (*Jeronimo and Robert, 2014*). In *kin28-AA* yeast lacking the Mediator tail module triad, Med17 ChIP signal was reduced much more than the reduction in Mediator occupancy (less than two-fold) reported for *med3Δ med15Δ* yeast (*Jeronimo et al., 2016*). Nonetheless, Med17 occupancy was still detectable at the same promoters, while an untagged control normalized to input yielded no signal (*Figure 1C–D*; *Figure 1—figure supplement 1C*). The reduction in Med17 occupancy was proportionally larger at SAGA-dominated genes than at TFIID-dominated genes, possibly reflecting the enrichment of SAGA-dominated genes among genes depending most strongly on the tail module triad for transcription, but association with both categories of gene promoters was generally reduced (*Figure 1C–D*) (*Ansari et al., 2012a*).

We also examined Med17 occupancy for a set of 498 genes for which Mediator association was detected at UAS regions in *KIN28*-WT yeast, which we refer to as 'UAS genes' (*Jeronimo et al., 2016*) (*Figure 1C*, right panel). Med17 occupancy is readily visible at promoters and UAS regions of these genes in *kin28-AA* yeast having an intact tail module triad (*Figure 1C*, upper right panel). Mediator head module subunits exhibit weaker ChIP signal at many UAS regions than do tail module subunits (*Jeronimo et al., 2016*; *Petrenko et al., 2016*); nonetheless, Mediator occupancy was evident at many UAS regions, as seen in the heat map and the shoulder on the upstream side of the line graph in *Figure 1C*, upper right panel. This occupancy appears to be reduced to near baseline levels in *med2Δ med3Δ med15Δ* yeast, as was reported previously for *med3Δ med15Δ* yeast (*Jeronimo et al., 2016*; *Petrenko et al., 2016*).

Despite a considerable decrease in Mediator occupancy at gene promoters as measured by ChIP in *med2Δ med3Δ med15Δ* yeast, only modest effects were observed on growth and transcript levels. Previous work has uncovered several examples in which decreased transcription caused by loss or impairment of general transcription factors, including Pol II, is compensated for by increased mRNA stability (*Baptista et al., 2017*; *Sun et al., 2012*; *Warfield et al., 2017*). Therefore, to investigate more directly the effect of decreased Mediator recruitment on transcription, we examined Pol II occupancy in wild type and *med2Δ med3Δ med15Δ* yeast by ChIP-seq. Wild type and *med2Δ med3Δ med15Δ* yeast whole cell extracts were spiked with extracts from *S. pombe* prior to immunoprecipitation with a monoclonal antibody to Rpb1, the largest subunit of Pol II, to allow global changes in Pol II occupancy to be measured. We observed an approximately four-fold decrease in Pol II occupancy in *med2Δ med3Δ med15Δ* yeast, while the normalized spike-in control showed equal IP efficiency in the two samples (*Figure 2*). Two features of this data are notable: first, both Mediator and Pol II occupancy are reduced at essentially all genes in *med2Δ med3Δ med15Δ* yeast; and second, substantial occupancy by Mediator and Pol II is observed even in the complete absence of the tail module triad. Thus, the tail module triad contributes broadly if not universally to recruitment of Mediator and Pol II, but at the same time is not absolutely required at the large majority of mRNA genes.

## Mediator association at UAS regions is stabilized by loss of TBP or Pol II

Previous work had indicated, and results of *Figure 1* demonstrate unequivocally, that Mediator can associate with many promoters independently of the tail module triad, albeit at lower occupancy

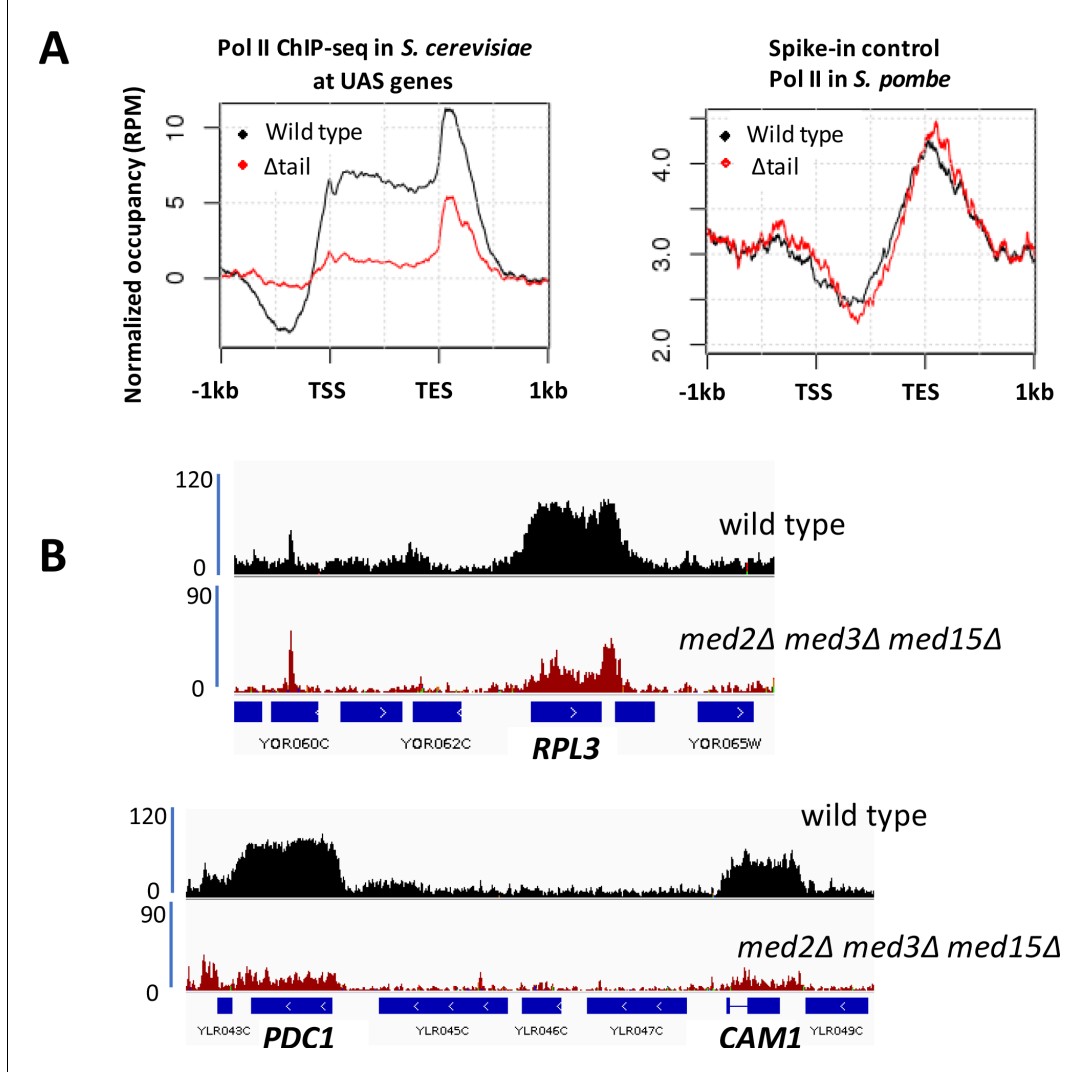

**Figure 2.** Decreased Pol II occupancy in *med2Δ med3Δ med15Δ* yeast. (**A**) ChIP-seq of Rpb1 in wild type (BY4741) and *med2Δ med3Δ med15Δ* (Δtail) yeast. Left panel: normalized occupancy at *S. cerevisiae* UAS genes, defined as in *Figure 1*. Right panel: normalized occupancy over 1150 *s. pombe* genes having expression levels > 1 (*Shetty et al., 2017*). (For unknown reasons, ChIP-seq against Rpb1 in *S. pombe* exhibits low signal at 5' regions and stronger signal at and beyond 3' regions of transcribed genes; see for example (*Shetty et al., 2017*; *Lee et al., 2017*)). (**B**) Browser scans showing normalized Rpb1 occupancy in wild type (BY4741) and *med2Δ med3Δ med15Δ* yeast.

DOI: https://doi.org/10.7554/eLife.39633.004

(*Jeronimo et al., 2016*; *Petrenko et al., 2016*). Based on known interactions of Mediator head and middle subunits with the general transcription machinery (*Plaschka et al., 2015*; *Plaschka et al., 2016*; *Ansari and Morse, 2013*; *Soutourina et al., 2011*; *Eychenne et al., 2016*; *Esnault et al., 2008*; *Nozawa et al., 2017*; *Tsai et al., 2017*; *Robinson et al., 2016*), we had previously suggested that Mediator might be recruited to tail module-independent targets via interactions with PIC components (*Ansari et al., 2012a*; *Ansari and Morse, 2012b*). To examine this possibility, we monitored Mediator occupancy genome-wide before and after depletion of PIC components using the anchor away method (*Haruki et al., 2008*).

We chose three PIC components to subject to depletion: TATA-binding protein (TBP); Rpb3, an essential subunit of Pol II; and Taf1, a TFIID subunit that is not shared with the SAGA complex (*Bhaumik, 2011*). Depletion of any of these PIC components resulted in loss of viability, as expected (*Figure 3—figure supplement 1A–B*) (*Haruki et al., 2008*; *Petrenko et al., 2017*; *Warfield et al., 2017*). Pilot experiments showed that treatment of *tbp-AA* yeast with rapamycin resulted in

depletion of TBP to near background levels, as measured by ChIP, within 30–60 min, consistent with prior work (*Figure 3—figure supplement 2A*) (*Haruki et al., 2008*; *Wong et al., 2014*; *Tramantano et al., 2016*; *Grimaldi et al., 2014*). Previous studies using anchor away of components of the transcription machinery have employed rapamycin treatment times varying from 30 min to 2 hr (*Anandhakumar et al., 2016*; *Baptista et al., 2017*; *Haruki et al., 2008*; *Jeronimo et al., 2016*; *Petrenko et al., 2016*; *Petrenko et al., 2017*; *Tramantano et al., 2016*; *Warfield et al., 2017*; *Wong et al., 2014*). Based on these studies and our results on TBP depletion, we chose to use one hr of rapamycin treatment as a time likely to allow a reasonable balance between attaining thorough depletion while also keeping indirect effects to a minimum.

We first examined Mediator binding before and after depletion of TBP, Rpb3, and Taf1 using myc-tagged Med15, from the tail module triad, and Med18, from the head module. As mentioned earlier, Mediator ChIP signal at promoters is generally low due to rapid turnover during transcription, and many active promoters do not exhibit detectable Mediator ChIP signal at UAS sites (*Jeronimo and Robert, 2014*; *Wong et al., 2014*). We therefore narrowed our examination of Mediator occupancy to the 498 'UAS genes' that exhibit Mediator occupancy at UAS regions in wild type yeast (*Jeronimo et al., 2016*). In agreement with prior reports, we observed association of both Med15 and Med18 at UAS regions for most of these genes, while a shift to the promoter region was seen for both Mediator subunits in *kin28-AA* yeast after rapamycin treatment (*Figure 3A*).

We had anticipated that depletion of PIC components might, by reducing interactions with Mediator at promoter regions, reduce Mediator occupancy at both promoter and UAS regions. However, Med15 and Med18 occupancy at UAS regions did not decrease upon depletion of TBP, Rpb3, or Taf1 (*Figure 3A–B*). For Med15, high occupancy was observed at UAS regions upon depletion of all three PIC components; occupancy was increased relative to that seen before depletion for TBP and Rpb3, but was constitutively high in the *taf1-AA* strain. This behavior was also evident at individual promoters, as Med15 occupancy increased at UAS sites upon depletion of TBP or Rpb3 but was constitutively high in *taf1-AA* yeast (*Figure 3C*). Consistent with these results, comparison of Med15 ChIP signal at UAS genes in FRB-tagged anchor away strains with an untagged control indicated that no effect of the epitope tag was seen in *rpb3-AA* yeast in the absence of rapamycin, while a modest increase (about 1.5 to 2-fold) was observed in *tbp-AA* yeast. A greater increase in signal was observed in *taf1-AA* yeast (*Figure 3—figure supplement 2B*); possibly the FRB tag in the latter strain interferes in some way with PIC assembly or function. For Med18, the ChIP signal was unchanged by depletion of TBP, Rpb3, or Taf1 (*Figure 3B*). ChIP-seq confirmed that all three PIC components were depleted, although Taf1 depletion was somewhat less efficient (*Figure 3—figure supplements 3A–C and and 4A*). Depletion of TBP essentially eliminated Pol II association, as expected, while depletion of Pol II resulted in a moderate reduction in TBP association, consistent with recently published work (*Figure 3—figure supplement 3D*) (*Joo et al., 2017*). ChIP-seq before and after depletion of TBP and Rpb3 against another tail module triad subunit, Med2, and against the scaffold subunit Med14 yielded results most similar to those seen with Med15, with both subunits showing increased occupancy after depletion of TBP or Rpb3 (*Figure 3D*).

The increased ChIP signal for Med15, Med2, and Med14 observed at UAS regions upon depletion of TBP or Rpb3 suggested that PIC component depletion might result in Mediator peaks being observed at UAS regions where they are normally difficult to detect. To test this idea, we examined Med15 ChIP-seq signal over all genes having transcription level >32 (*Nagalakshmi et al., 2008*), after filtering out the 498 genes identified as exhibiting Mediator UAS peaks, leaving about 1400 genes (*Figure 4A*). Indeed, this analysis revealed several hundred genes that showed increased Mediator occupancy in their UAS regions upon TBP depletion where negligible occupancy was observed by ourselves or others under normal growth conditions (*Jeronimo et al., 2016*). Gene ontology (GO) analysis of the 200 genes from this group showing the greatest increase in Med15 occupancy revealed a strong enrichment for ribosomal protein (RP) genes, and a direct examination of RP genes showed increased Mediator occupancy at UAS regions upon depletion of TBP (*Figure 4B and C*). Promoters of some non-RP, 'non-UAS' gene promoters also showed increased Mediator occupancy upon depletion of TBP; an example is shown in *Figure 4C*. Taken together, the results of *Figures 3–4* indicate that Mediator association with UAS regions does not decrease upon depletion of PIC components, and that association apparently increases upon depletion of TBP or Pol II.

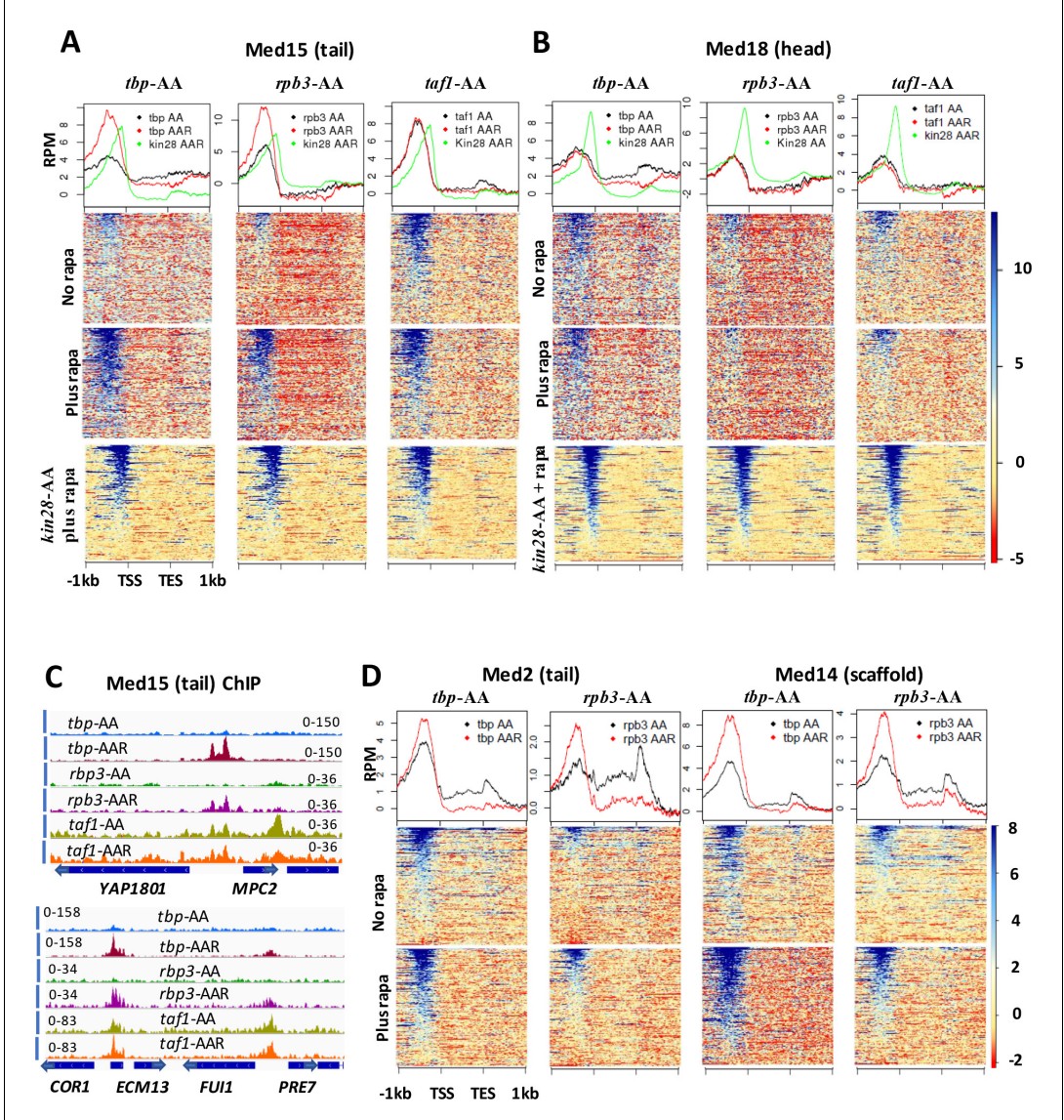

**Figure 3.** Effect of depletion of TBP and Rpb3 on Mediator occupancy. ChIP-seq was performed against (**A**) Med15 and (**B**) Med18 in *tbp*-AA, *rpb3*-AA, and *taf1*-AA yeast with ('AAR') and without ('AA') rapamycin treatment, and in *kin28*-AA yeast after rapamycin treatment. Heat maps and line graphs (normalized reads per million, RPM) depict results combined from two independent experiments (except for Med18 in *taf1-AA* yeast) averaged over 498 genes exhibiting detectable Mediator ChIP signal at UAS regions in wild type yeast (*Jeronimo et al., 2016*). (**C**) Browser scans showing occupancy of Med15 at two chromosomal regions in *tbp*-AA, *rpb3*-AA, and *taf1*-AA yeast with and without rapamycin treatment. *YBL044W*, between *COR7* and *ECM13*, is an uncharacterized ORF and likely contains a UAS, as has been observed at many other uncharacterized or dubious ORFs (*Paul et al., 2015*). (**D**) Med2 and Med14 occupancy measured by ChIP-seq with and without depletion of TBP or Rpb3, at UAS genes.

DOI: https://doi.org/10.7554/eLife.39633.005

The following figure supplements are available for figure 3:

**Figure supplement 1.** Loss of viability upon rapamycin treatment of anchor away strains used in this study.

DOI: https://doi.org/10.7554/eLife.39633.006

**Figure supplement 2.** Control experiments for PIC component depletion.

DOI: https://doi.org/10.7554/eLife.39633.007

**Figure supplement 3.** Effect of PIC component depletion on normalized occupancy (reads per million) by other PIC components, shown at UAS genes.

DOI: https://doi.org/10.7554/eLife.39633.008

**Figure supplement 4.** Browser scans showing depletion of PIC components and differential ChIP signal for Mediator tail and head module subunits at proximal promoter and UAS regions.

DOI: https://doi.org/10.7554/eLife.39633.009

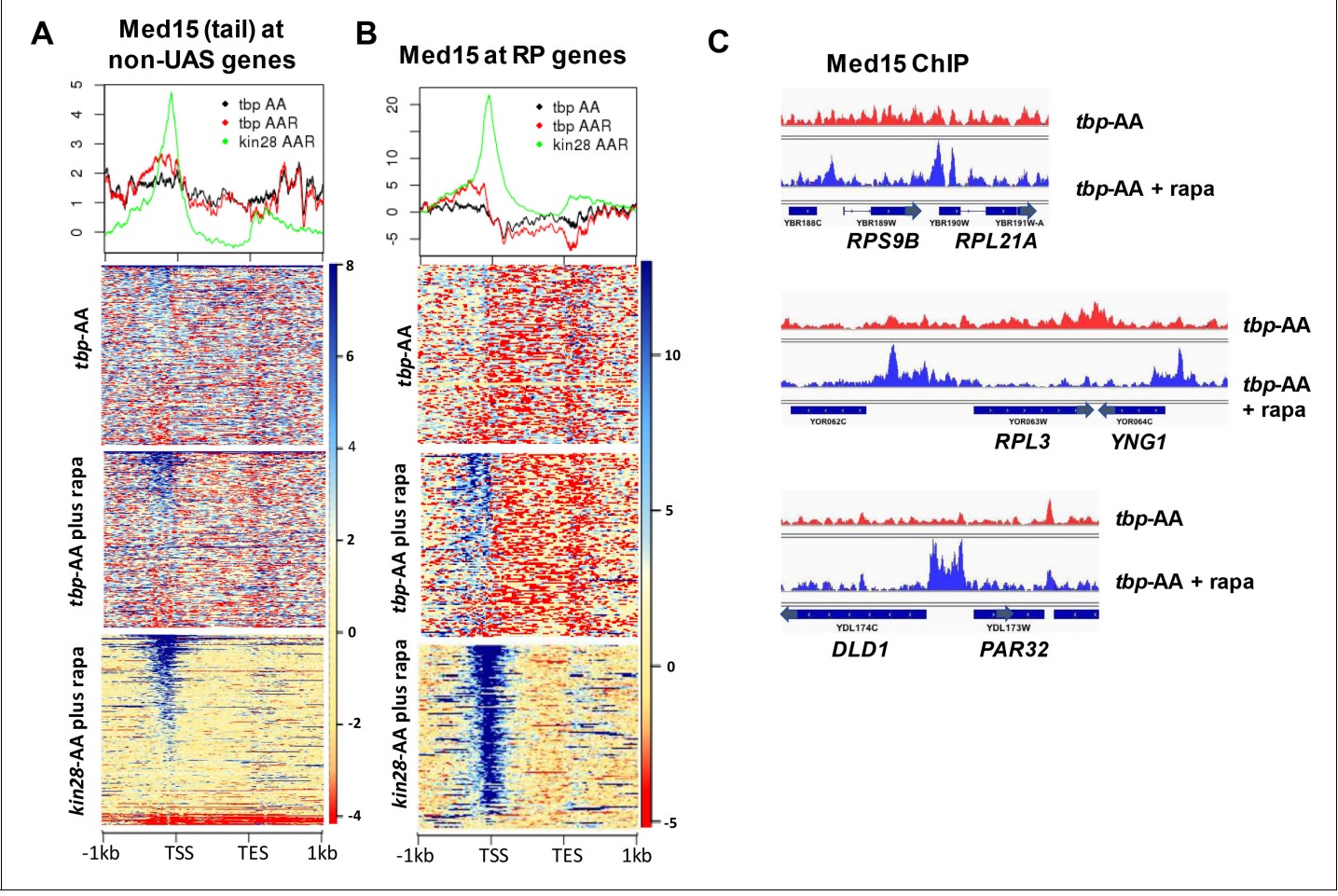

**Figure 4.** Effect of depletion of TBP on Mediator occupancy at 'non-UAS' genes. (**A**) Med15 occupancy measured by ChIP-seq at the 2000 most highly transcribed genes, after removal of the 498 genes exhibiting detectable Mediator ChIP signal at UAS regions in wild type yeast (*Jeronimo et al., 2016*), in *tbp*-AA yeast with and without rapamycin treatment. Occupancy after depletion of Kin28 (bottom) is shown for reference. (**B**) Med15 occupancy measured by ChIP-seq at ribosomal protein (RP) genes and at the *DLD1-PAR32* promoter in *tbp*-AA yeast with and without rapamycin treatment.
DOI: https://doi.org/10.7554/eLife.39633.010

A possible explanation for increased occupancy of Mediator subunits at UAS regions upon depletion of TBP or Rpb3 is that interruption of the transcription cycle by PIC disruption prevents the transit of Mediator from UAS regions to promoters and off the DNA. At genes for which Mediator association with UAS regions is observed, a single Mediator complex bridges the UAS and the promoter, with tail module subunits contacting the UAS and head and middle subunits engaging the promoter region through their contacts with Pol II and GTFs (*Jeronimo et al., 2016*; *Petrenko et al., 2016*). Interruption of the latter contacts could strand Mediator at the UAS. It is not clear why an increase in ChIP signal is observed for Med15, but not for Med18. One possibility is that the tail module, which can associate with UAS regions independently of the rest of Mediator under some conditions (*Ansari et al., 2009*; *He et al., 2008*; *Paul et al., 2015*; *Zhang et al., 2004*), may bind to UAS regions independently of the head module at least some of the time (*Anandhakumar et al., 2016*). Arguing against this explanation is the observation that Med14, which contacts subunits from Mediator head, middle, and tail modules, shows increased occupancy upon depletion of TBP or Rpb3, thus behaving similarly to Med15 and Med2 subunits from the tail module triad. It is also possible that Mediator may adopt an altered configuration under these conditions, such that immunoprecipitation of Med18 is less efficient upon depletion of TBP or Rpb3. In this regard, it is notable that under conditions of Kin28 depletion, ChIP signal for Med15 is observed at UAS regions of some genes while subunits from the head module do not give detectable signal,

while ChIP signals are observed for both Med15 and head module subunits at proximal promoter regions (*Figure 3—figure supplement 4B*) (*Jeronimo et al., 2016*).

Although the notion of Mediator transit from UAS to promoter depending on PIC integrity is consistent with recent models of Mediator function at UAS regions (*Jeronimo et al., 2016*; *Petrenko et al., 2016*), the results so far do not help clarify how Mediator is recruited to promoters at the many genes for which Mediator association is not readily observed at upstream regions, or how recruitment occurs in the absence of the Mediator tail module triad. We therefore next modified our approach to allow PIC component depletion under conditions in which Mediator association with gene promoters was stabilized.

## Mediator association with promoters depends on PIC components

We next asked how PIC component depletion affected Mediator association under conditions in which Mediator binding to proximal gene promoters was stabilized. For this purpose, we constructed double anchor away strains in which Kin28 and either TBP, Taf1, or Rpb3 were tagged with FRB to allow nuclear depletion upon treatment with rapamycin. We then separately introduced Myc-tagged Med15 and Med18 into each double anchor away strain. Depletion of Kin28 alone allows detection by ChIP of Mediator occupancy at promoters (*Jeronimo and Robert, 2014*; *Wong et al., 2014*); comparison with the double depletion strains after rapamycin treatment then allows determination of the effect of depleting TBP, Taf1, and Rpb3 on Mediator occupancy at gene promoters. As expected, these strains were all sensitive to rapamycin (*Figure 3—figure supplement 1*). We note that the *kin28-tbp-AA* strain harboring the Med15-myc epitope was slow growing; the *med15-myc* allele has been previously been found to behave as a hypomorph in conjuction with a *med3Δ* mutation (*Ansari et al., 2012a*; *Zhang et al., 2004*), and it evidently exhibits a negative growth phenotype in the context of the *kin28-tbp-AA* strain as well.

We first examined the effect of depleting PIC components on Mediator occupancy at all genes, separated into SAGA-dominated and TFIID- dominated genes (*Huisinga and Pugh, 2004*). Our rationale was predicated on findings that genes that depend on the tail module triad for transcription are enriched for SAGA- dominated genes, relative to TFIID- dominated genes (*Ansari et al., 2012a*), and that loss of the tail module caused a more pronounced decrease in Mediator occupancy at promoters of SAGA-dominated genes than at TFIID-dominated genes (*Figure 1*). Based on these observations, we hypothesized that TFIID-dominated genes might show greater dependence than SAGA-dominated genes on PIC components for Mediator occupancy.

Depletion of either Taf1 or Rpb3 resulted in decreased occupancy of Med15 and Med18 in comparison to depletion of Kin28 alone at almost all TFIID-dominated genes (*Figure 5A–D*). Decreased Med15 and Med18 occupancy was also observed at SAGA-dominated genes and UAS genes upon depletion of Rpb3, along with a partial shift towards the UAS region, while depletion of Taf1 also resulted in decreased Med18 occupancy but little change in Med15 occupancy, other than a small upstream shift, at SAGA-dominated and UAS genes. A more pronounced decrease in Med15 or Med18 occupancy at TFIID-dominated genes compared to SAGA-dominated genes after depletion of Rpb3 or Taf1 could also be observed at genes having similar levels of transcription (*Figure 5—figure supplement 1*), and examples can also be found of individual TFIID-dominated genes exhibiting greater loss of Mediator ChIP signal than at SAGA-dominated genes (*Figure 5—figure supplement 2*). We conclude that under conditions of Kin28 depletion, Mediator occupancy at promoters of TFIID-dominated genes, and to lesser extent SAGA-regulated genes, depends on both Taf1 and Pol II.

## TBP is required for transit of Mediator from UAS to promoter

The effect of depleting TBP together with Kin28 was markedly different from that of depleting Taf1 or Rpb3 and Kin28 (*Figure 5A–D*). At SAGA-dominated genes, occupancy of Med15 from the Mediator tail module did not decrease and shifted upstream in *kin28-tbp-AA* yeast compared to *kin28-AA* yeast (*Figure 5A & C*). This shift is also evident at UAS genes (*Figure 5C*), and yields a profile nearly identical to that seen upon depletion of TBP alone (*Figure 5—figure supplement 3*), suggesting that Mediator transit from UAS to promoter was inhibited by depletion of TBP. At TFIID-dominated genes, depletion of TBP resulted in an upstream shift of smaller magnitude, along with a reduction in the Med15 ChIP signal (*Figure 5A & C*). The smaller shift seen for TFIID-dominated

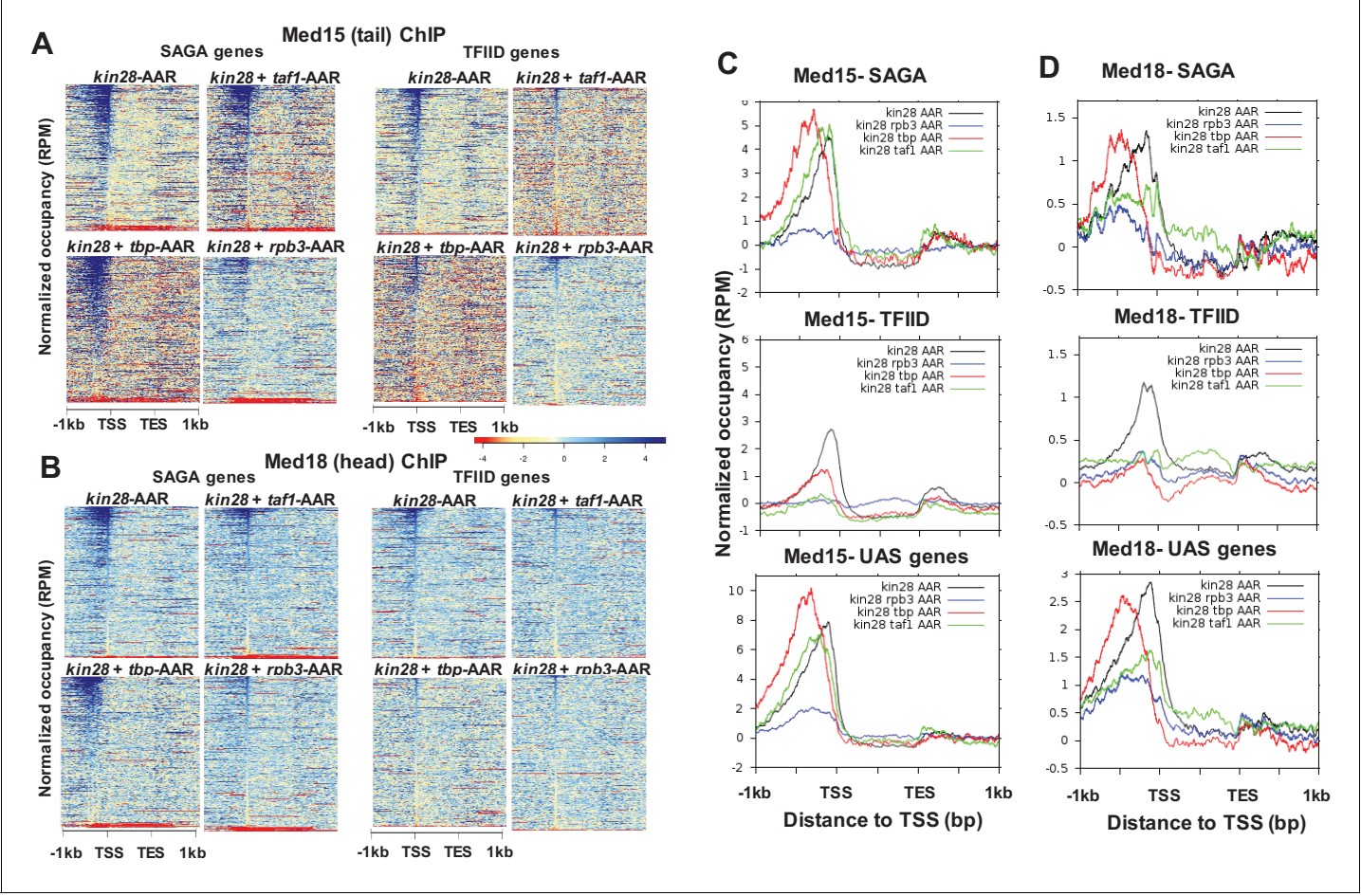

**Figure 5.** Effect of depletion of PIC components together with Kin28 on Mediator occupancy. (**A**) Heat maps showing occupancy of Med15 after depletion of Kin28 alone, or in combination with Taf1, TBP, or Rpb3, at SAGA-dominated and TFIID-dominated genes. (**B**) Same as (**A**), for Med18 occupancy. (**C**) Line graphs (normalized reads per million) depicting Med15 occupancy after depletion as in (**A**) and (**B**), aligned at the TSS and shown for SAGA-dominated (top), TFIID-dominated (middle), and UAS (bottom) genes. (**D**) Same as (**C**), but for Med18. Combined data for two replicate experiments was used for *kin28-tbp-AA* yeast; all other plots derive from single experiments that were consistent with replicate experiments. We note that variable results were observed for the effect on Med15 occupancy in *kin28-taf1-AA* yeast at SAGA-dominated genes, with peak intensity being reduced in some experiments and nearly unchanged in others, while reduced occupancy was consistently observed at TFIID genes.
DOI: https://doi.org/10.7554/eLife.39633.011

The following figure supplements are available for figure 5:

**Figure supplement 1.** Effect of depletion of PIC components together with Kin28 on Mediator occupancy at similarly expressed SAGA-dominated and TFIID-dominated genes.
DOI: https://doi.org/10.7554/eLife.39633.012

**Figure supplement 2.** Browser scans of normalized Mediator occupancy after simultaneous depletion of Kin28 and Rpb3 or Kin28 and Taf1.
DOI: https://doi.org/10.7554/eLife.39633.013

**Figure supplement 3.** Med15 occupancy at UAS regions after depletion of TBP, Kin28, or Kin28 and TBP together.
DOI: https://doi.org/10.7554/eLife.39633.014

**Figure supplement 4.** Normalized occupancy of Med2 and Med17 at SAGA and TFIID genes after depletion of Kin28 and TBP together compared to Kin28 alone
DOI: https://doi.org/10.7554/eLife.39633.015

genes than for SAGA-dominated genes is consistent with the average distance between UAS and TSS being larger for TATA-containing than for TATA-less promoters (*Erb and van Nimwegen, 2011*; *Kristiansson et al., 2009*), supporting the notion that depletion of TBP causes Mediator to be stranded at the UAS.

Med18, from the head module, as well as Med2 from the tail module triad and Med17 from the head module all showed a shift upstream at SAGA-dominated genes upon TBP depletion, but the signals for these subunits were reduced to varying degrees upon depletion of Kin28 and TBP compared to those observed upon depletion of Kin28 alone (*Figure 5* and *Figure 5—figure supplement 4*). At TFIID-dominated genes, depletion of TBP and Kin28 resulted in a nearly complete loss of ChIP signal for Med18, Med2, and Med17 (*Figure 5B and D* and *Figure 5—figure supplement 4*). Examination of individual gene loci corroborates these observations, with Mediator occupancy shifting upstream at SAGA-dominated genes and often disappearing at TFIID-dominated genes upon TBP depletion (*Figure 6*).

Depletion of Rpb3 together with Kin28 strongly decreased occupancy of both Med15 and Med18, especially at TFIID-dominated genes (*Figure 5*). In addition, a shift in Med15 occupancy to more upstream regions (together with an overall decrease in occupancy) was seen at SAGA-dominated genes upon depletion of Rpb3 and Kin28 (*Figure 5A & C*). This molecular phenotype is also apparent upon analysis of the 498 genes at which Mediator occupancy is observed in wild type yeast (*Figure 5C*) (*Jeronimo et al., 2016*).

The effects seen in *kin28-rpb3-AA* yeast may be due to a direct effect on Mediator transit to and association with promoters, or may reflect the impact of loss of Pol II on the cooperative assembly of the PIC. Depletion of Pol II results in partial loss of TBP (*Figure 3—figure supplement 3D*) (*Ansari et al., 2014*; *Joo et al., 2017*; *Sharma et al., 2003*). TBP-containing preinitiation complexes that are present, but which lack Pol II, may facilitate transit of Mediator from the UAS to the promoter, but association may be destabilized by the absence of normal contacts between Mediator and Pol II (*Soutourina et al., 2011*) or other PIC components that are less stably bound in the absence of Pol II. The combination of these effects could produce the effects seen upon depletion of Kin28 and Rpb3. This differential dependence on Rpb3 and TBP for Mediator transit may pertain

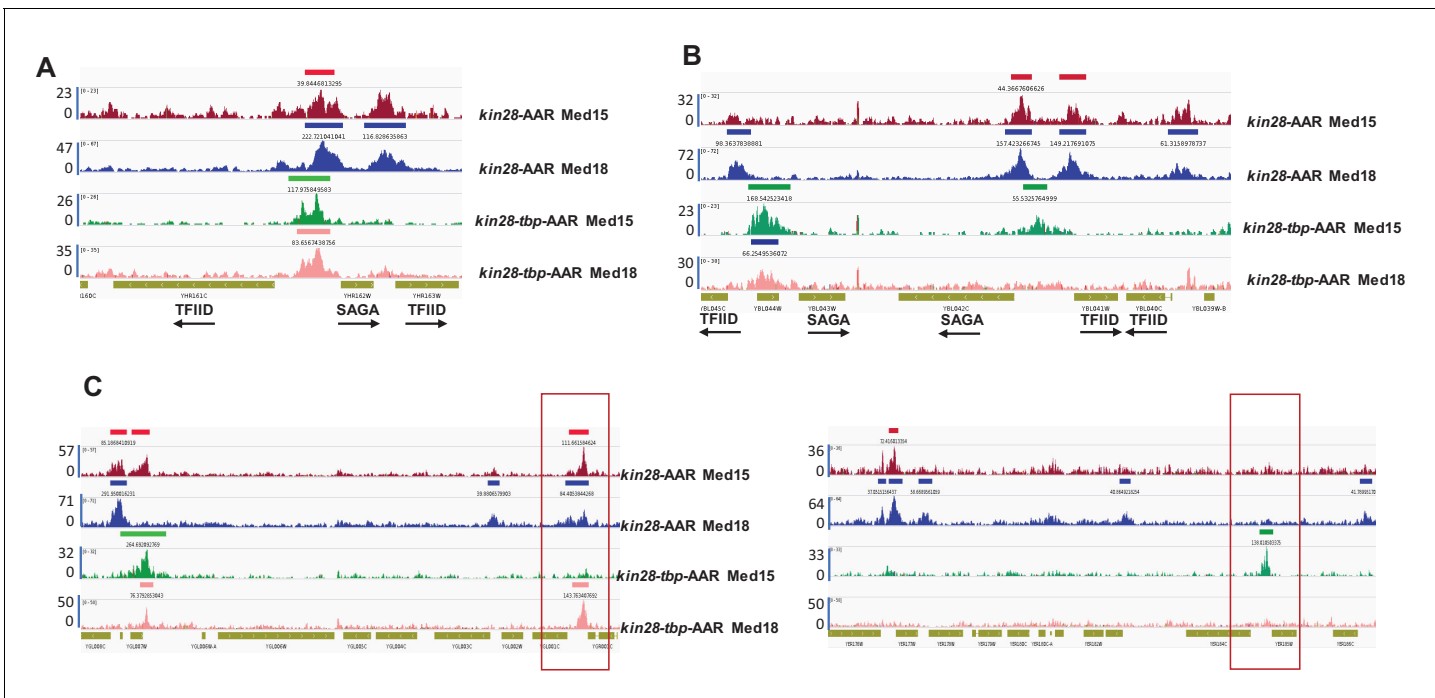

**Figure 6.** TBP depletion shifts Mediator occupancy from promoter to UAS sites. Browser scans showing normalized Med15 and Med18 occupancy in *kin28*-AA and *kin28-tbp*-AA yeast treated with rapamycin, as indicated. (**A**) Upstream shift of Mediator at a SAGA-dominated gene and loss of occupancy at a TFIID-dominated gene upon depletion of TBP. Arrows depict direction of transcription. (**B**) Loss of Mediator peaks at promoters and gain of Mediator peaks at upstream regions at both TFIID and SAGA-dominated genes. *YBL044W* is an uncharacterized ORF and likely contains a UAS, as has been observed at many other uncharacterized or dubious ORFs (*Paul et al., 2015*). (**C**) At some loci, new Mediator peaks are visible after dual depletion of Kin28 and TBP (boxed regions).
DOI: https://doi.org/10.7554/eLife.39633.016

only under the special conditions of Kin28 depletion, as increased Mediator ChIP signal was observed at UAS regions when either TBP or Rpb3 was depleted when Kin28 activity was unperturbed (*Figure 3*).

The cooperative nature of PIC assembly must be taken into account when interpreting results of depletion of individual PIC components. Depletion of TBP abolishes recruitment of Pol II (*Figure 3—figure supplement 3D*), and this is also the case when Kin28 is simultaneously depleted (*Figure 7A*). These results, in conjunction with the upstream shift in Mediator occupancy upon depletion of Kin28 and TBP compared to depletion of Kin28 alone (*Figure 5*), allow us to infer that TBP is necessary for Mediator transit to promoter regions under conditions of Kin28 depletion, but not whether it is sufficient, as Pol II may contribute. In addition, since Taf1 association is only slightly decreased upon depletion of TBP by itself or simultaneously with Kin28 (*Figure 7B*), we infer that Taf1 is not sufficient for Mediator transit from UAS to promoter.

With regard to the role of Taf1 and Pol II in stable occupancy of Mediator at gene promoters (*Figure 5*), depletion of Rpb3 results in a modest (less than two-fold) reduction in association of both TBP and Taf1 (*Figure 3—figure supplement 3D* and *Figure 7C*). In contrast, depletion of Taf1 has been shown to reduce Pol II occupancy on average about 4-fold, with little difference between SAGA-dominated and TFIID-dominated genes (*Warfield et al., 2017*). However, Taf1 (and presumably TFIID) by itself appears not to be sufficient to retain Mediator at gene promoters after Kin28

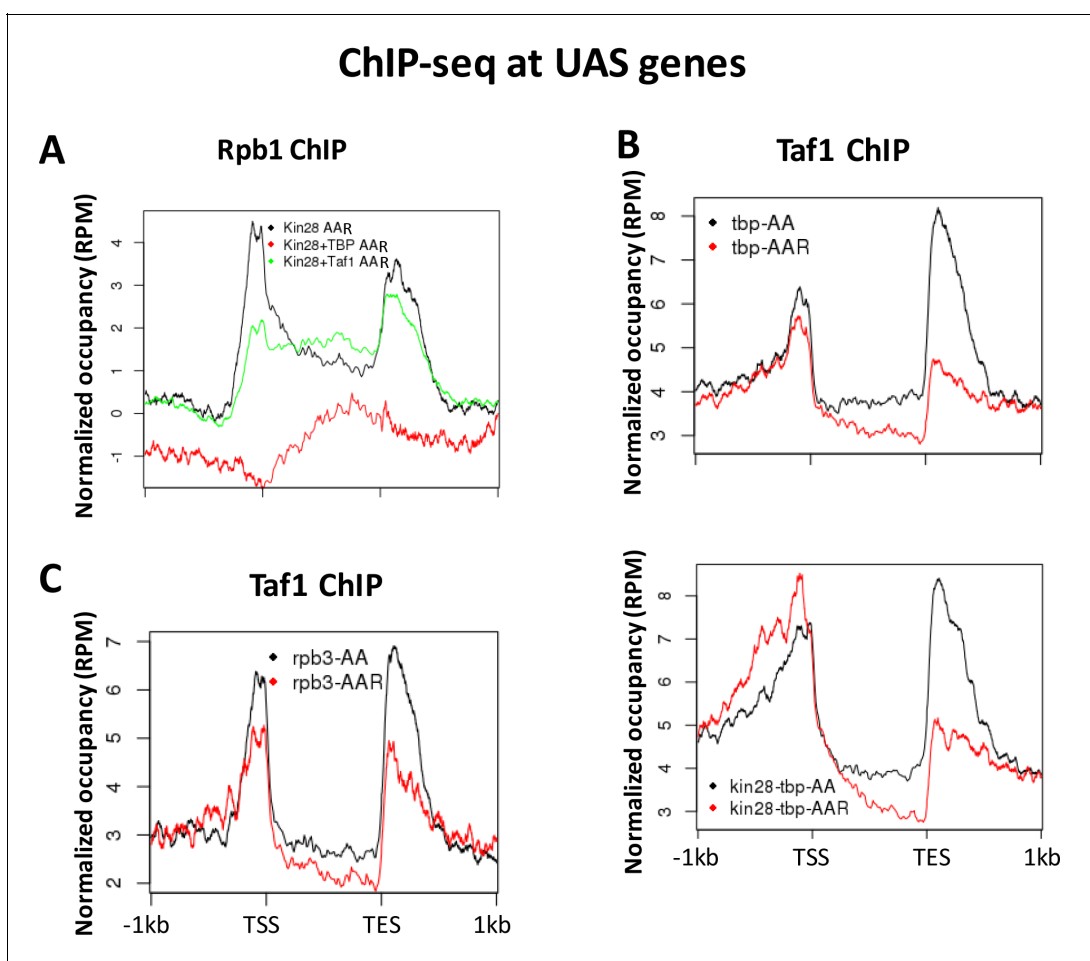

**Figure 7.** Normalized PIC occupancy at UAS genes after PIC component depletion. (**A**) Rbp1 occupancy after depletion of Kin28 (black), Kin28 and TBP (red), and Kin28 and Taf1 (green). (**B**) Taf1 occupancy before (black) and after (red) depletion of TBP (top graph) or TBP and Kin28 (bottom graph). (**C**) Taf1 occupancy before (black) and after (red) depletion of Rpb3. The prominent peaks observed near the TES appear to represent artifactual ChIP signal sometimes seen at 3' regions of highly transcribed genes; similar signals are observed in ChIP against GFP expressed in yeast (*Paul et al., 2015*).
DOI: https://doi.org/10.7554/eLife.39633.017

depletion, as Taf1 is retained when Rpb3 is depleted, while Mediator occupancy is essentially abolished. Evidently the partial loss of Pol II accompanying depletion of Taf1 is not sufficient to cause appreciable loss of TBP, as no upstream shift in Mediator occupancy was observed upon depletion of Taf1 and Kin28. The alternative possibility that Taf1 is required for recruitment of Mediator to UAS regions is ruled out by our observation that Mediator occupancy of UAS regions was not reduced upon depletion of Taf1 alone (*Figure 3*). Finally, it appears that the level of Pol II still associated with gene promoters after Taf1 depletion is sufficient to allow Mediator to stably associate, at least at SAGA-dominated genes (compare effects on Med15 occupancy in *kin28-rpb3-AA* and in *kin28-taf1-AA* yeast (*Figure 5A & C*).

In summary, our results indicate that TBP, which is critical for PIC formation, is required for transit of Mediator from UAS to gene promoters, while partially formed PICs present after depletion of Taf1 or Rpb3 allow this transit but lead to unstable Mediator-PIC association at promoters under conditions that prevent the normal rapid turnover of Mediator that accompanies release of Pol II.

## Discussion

The major findings reported in this work are that 1) the Med2-Med3-Med15 tail module triad plays a major role in recruitment of Mediator and Pol II, but is dispensable for viability; 2) Mediator association with active gene promoters is observed even in the complete absence of the tail module triad; 3) PIC components, in particular Taf1 and Pol II (Rpb3) contribute to Mediator association with gene promoters, especially at TFIID-dominated genes, and 4) TBP plays a critical role in the transit of Mediator from UAS to promoter sites.

Previous studies reported altered transcript levels for only 5–10% of all genes in mutants lacking one or two tail module triad subunits, suggesting that this moiety might serve as a requisite target for transcriptional activators only for a limited fraction of the genome (*Ansari et al., 2012a*; *van de Peppel et al., 2005*). Pol II occupancy was only moderately reduced in *med3Δ med15Δ* yeast, particularly at TFIID-dominated genes, and Mediator occupancy at gene promoters under conditions of Kin28 inactivation was reduced by less than two-fold in this same mutant, further suggesting a limited role for the tail module triad in recruitment of Mediator and Pol II (*Jeronimo et al., 2016*; *Paul et al., 2015*). In the present work, however, we find a more substantial reduction of occupancy by Mediator and Pol II in *med2Δ med3Δ med15Δ* yeast, indicating a widespread role for the tail module triad in recruitment of the Mediator complex and functional redundancy in the tail module subunits, reminiscent of previous observations in comparing *med3Δ med15Δ* yeast to single deletion mutants (*Ansari et al., 2012a*). This redundancy could reflect 'fuzzy' interactions between activators and subunits of the tail module triad, such that loss of any individual tail module triad subunit would result in loss but not elimination of recruitment and consequent activation (*Tuttle et al., 2018*).

Recruitment of Mediator via the tail module triad is likely to occur via activators bound to UAS sites, as in the canonical model for gene activation (*Figure 8A*). Consistent with this model is the finding that Mediator is detected upstream of active genes by ChEC-seq, at distances consistent with occupancy of UAS sites, in both the presence and absence of functional Kin28 (*Grünberg et al., 2016*; *Grünberg and Zentner, 2017*). This argument is further supported by our findings that Mediator occupancy is observed by ChIP at upstream regions of genes when both TBP and Kin28 are depleted (*Figure 5*). Altogether, then, these various observations support a model in which Mediator is principally recruited first to UAS sites by activators contacting subunits of the tail module triad before transiting to the promoter and facilitating PIC formation.

Although the dominant mechanism of Mediator recruitment is likely to occur via tail-module dependent interactions with UAS-bound activators, our findings that Mediator and Pol II are recruited to gene promoters even in the complete absence of the tail module triad indicate the existence of an alternative, tail-module-independent mechanism for Mediator recruitment (*Figure 8B*). In this second pathway, PIC components could be recruited before or together with Mediator, presumably via activator-mediated interactions, and Mediator association would be facilitated by contacts with PIC components and in turn stabilize binding of those components (*Figure 8B*). In both pathways, following formation of the Mediator-PIC complex, TFIIH is recruited and its Kin28 subunit phosphorylates the carboxyl-terminal domain of the Rpb1 subunit of Pol II; this allows escape of Pol II into the elongation complex, and results in rapid loss of Mediator association (*Jeronimo and Robert, 2014*; *Wong et al., 2014*). Direct recruitment of Mediator to promoter regions via interactions

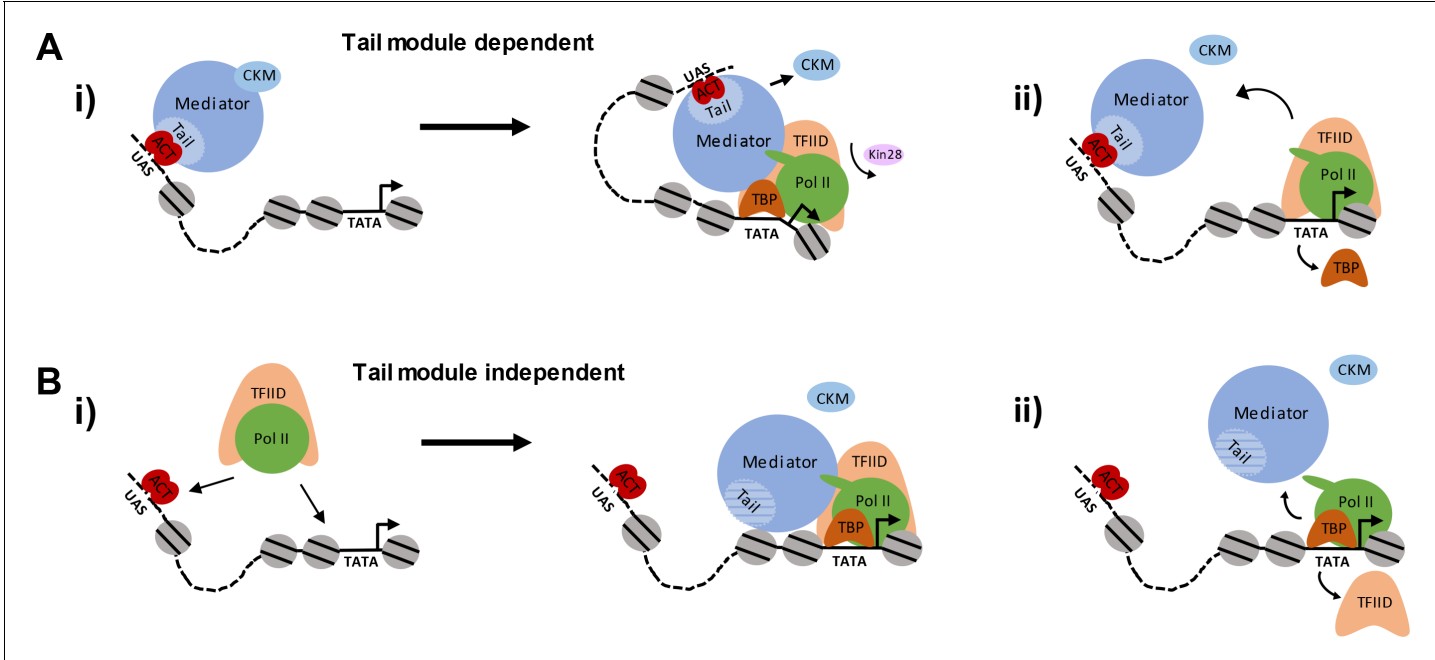

**Figure 8.** Pathways of Mediator recruitment. (**A**) (**i**) Mediator recruitment via interactions between activator proteins bound to UASs and the Med2-Med3-Med15 triad. Contacts between Mediator and components of the general transcription machinery facilitate transit of Mediator to the proximal promoter region, and stabilize association of both Mediator and GTFs. (ii) TBP plays a critical role in this transition, and its depletion results in Mediator being stranded at the UAS. (**B**) (**i**) Mediator recruitment via interactions between the middle/head modules and components of the PIC. In this pathway, the PIC is first recruited via interactions between UAS-bound activators and PIC components, such as Tafs within TFIID. (ii) Depletion of PIC components destabilizes association of Mediator with the proximal promoter, particularly at TATA-less, TFIID-dominated promoters.

DOI: https://doi.org/10.7554/eLife.39633.018

with PIC components may be favored at TFIID-dominated genes relative to SAGA-dominated genes; this would be consistent with the greater effect on Mediator occupancy seen at the former compared to the latter class upon depletion of PIC components (*Figure 5*).

Our results indicate that PIC components are important for transit of Mediator from UAS to promoter, and for association with promoters following that transit. PIC components are not necessary for recruitment of Mediator to UAS sites, as their depletion does not lead to reduction in Mediator association (*Figure 3*). Depletion of Taf1 results in greatly diminished Mediator occupancy at TFIID-dominated promoters, and little effect at SAGA-dominated promoters, whereas depletion of TBP causes a shift in Mediator occupancy to UAS regions at both promoter categories, along with diminished occupancy at TFIID-dominated genes (*Figure 5*). Thus, Taf1, and presumably TFIID, is important for stabilizing Mediator occupancy at promoters of TFIID-dominated genes following its transit from the initial site of recruitment, whereas TBP is critical for the transit from UAS to promoter of both SAGA- and TFIID-dominated genes (*Figure 8*, right). Depletion of Rpb3 together with Kin28 causes a decrease in Mediator occupancy at promoters, but also results in a new, upstream peak of Mediator occupancy that is much lower in intensity than that seen upon depletion of TBP and Kin28 (*Figure 5C–D*). We interpret this as a combined effect of Pol II stabilizing Mediator at promoter regions, together with Pol II depletion causing a partial decrease in TBP occupancy, which causes Mediator occupancy to shift upstream in a fraction of the cells being sampled (*Figure 3—figure supplement 3D*) (*Joo et al., 2017*).

Related to our findings that PIC components facilitate Mediator association with active gene promoters, in vitro results have indicated cooperative interactions among Mediator, TFIID, and Pol II (*Johnson and Carey, 2003*; *Johnson et al., 2002*; *Baek et al., 2002*; *Takahashi et al., 2011*). In vivo evidence for such cooperative interactions, however, has been sparse. An 'activator bypass' experiment showed that direct recruitment of TFIIB, via tethering to the DNA-binding domain of an activator, resulted in recruitment of a functional PIC and the Mediator complex in yeast, pointing to

'backwards' recruitment of Mediator via PIC interactions (*Lacombe et al., 2013*). More recently, depletion of Taf1 was shown to result in a widespread reduction in association of head module subunit Med8 with active genes when mapped by ChEC-seq (*Grünberg et al., 2016*; *Grünberg and Zentner, 2017*). Neither of these in vivo studies, however, examined PIC-Mediator cooperativity at promoter regions, and the distinct effects of depleting different PIC components on Mediator association (*Figure 5*) could not have been predicted on the basis of previous work.

Several interactions between Mediator and TFIID components, and Mediator and Pol II, have been documented that could contribute to Mediator association with promoters. Structural studies have shown the head module subcomplexes Med8-Med18-Med20 and Med11-Med17-Med22 bind with TBP (*Cai et al., 2010*; *Larivière et al., 2006*). Disruption of the Med8-Med18-Med20 subcomplex in *med18Δ* or *med20Δ* yeast does not cause loss of viability, suggesting that multiple Mediator-TBP interactions are possible at the proximal promoter for initiation (*Ranish et al., 1999*). In addition, interaction between the tail subunits Med15 or Med16 with TFIID subunits Taf14 or Taf1 have been observed in yeast two-hybrid and mass spectrometry experiments (*Gavin et al., 2006*; *Gavin et al., 2002*; *Lim et al., 2007*). Interactions between Mediator and Pol II have also been demonstrated in both structural and genetic studies (*Davis et al., 2002*; *Esnault et al., 2008*; *Plaschka et al., 2015*; *Soutourina et al., 2011*; *Cai et al., 2010*; *Robinson et al., 2015*).

Interactions between Mediator and Pol II or TFIID appear more important for stabilizing Mediator occupancy at TFIID-dominated genes than for SAGA-dominated genes, based on the stronger effect on Mediator occupancy in *med2Δ med3Δ med15Δ* yeast at the latter class than the former (*Figure 1*), and the stronger effect seen at TFIID-dominated genes upon depletion of Taf1 or Rpb3 (*Figure 5*). The distinction of these two pathways in terms of dependence on PIC components is also suggested by recent work showing that disruption of a Mediator-TFIIB connection via *med10-ts* yeast differently reduces association of Mediator, TFIIB, and GTFs (*Eychenne et al., 2016*). TFIID-dominated genes exhibit a greater reduction in Mediator rather than GTFs, while SAGA-dominated genes exhibit a greater reduction in GTFs rather than Mediator (*Eychenne et al., 2016*). Furthermore, two recent studies report that Mediator in general, and not just tail module subunits, contributes more strongly to Pol II recruitment at SAGA-dominated than TFIID-dominated genes (*Bruzzone et al., 2018*; *Petrenko et al., 2017*).

What is the basis for the differential dependence of SAGA-dominated and TFIID-dominated genes on Mediator? The average distance between UAS and TSS is greater for TATA-containing than TATA-less genes (*Kristiansson et al., 2009*; *Erb and van Nimwegen, 2011*). UASs show diminished ability to activate transcription from a reporter gene as the distance between UAS and TATA box increases (*Dobi and Winston, 2007*), and Mediator mutations were found to affect the dependence of UAS activity on distance (*Reavey et al., 2015*). Thus, the greater UAS-TSS distance of TATA-containing gene promoters may cause a greater dependency on Mediator for activation.

In summary, the work here provides new insights into the role of the Mediator tail module triad and PIC components in Mediator recruitment and dynamics. Future work will be aimed at a fuller understanding of the mechanism of tail module-independent recruitment of Mediator, and of the differential dependence of SAGA-dominated and TFIID-dominated genes on PIC components for stable Mediator association at gene promoters.

## Materials and methods

**Key resources table**

| Reagent type (species) or resource | Designation | Source or reference | Identifiers | Additional information |
|---|---|---|---|---|
| Strain, strain background (*S. cerevisiae*) | yFR1321 | PMID: 24704787 | | Francois Robert (University of Montreal) |
| Strain, strain background (*S. cerevisiae*) | KHW127 | PMID: 24746699 | | Kevin Struhl (Harvard Medical School) |

*Continued on next page*

*Continued*

| Reagent type (species) or resource | Designation | Source or reference | Identifiers | Additional information |
|---|---|---|---|---|
| Strain, strain background (*S. cerevisiae*) | LS01 | PMID: 15254252 | | Alan Hinnebusch (National Institutes of Health) |
| Strain, strain background (*S. cerevisiae*) | LS10 | PMID: 15254252 | | Alan Hinnebusch (National Institutes of Health) |
| Strain, strain background (*S. pombe*) | FWP510 | PMID: 24100010 | | Fred Winston (Harvard Medical School) |
| Antibody | 8WG16 (Pol II) (mouse monoclonal) | Covance | Cat# 664906 | ChIP, 2 µg (1:150) |
| Antibody | Rpb3 (mouse monoclonal) | Neoclone | Cat# W0012 | ChIP, 1 µg (1:300) |
| Antibody | 9E10 (Myc) (mouse monoclonal) | Roche | Cat# 11667141001 | ChIP, 2 µg (1:150) |
| Antibody | 9E10 (Myc) (mouse monoclonal) | Sigma | Cat# 11667149001 | ChIP, 5 µg (1:300) |
| Antibody | anti-protein A (rabbit polyclonal) | Sigma | Cat# P3775 | ChIP, 2.5 µg (1:250) |
| Antibody | TBP (affinity-purified rabbit polyclonal) | Tony Weil, Vanderbilt University | NA | ChIP, 2.5 µg (1:150) |
| Antibody | 58C9 (TBP) (mouse monoclonal) | Abcam | Cat# ab61411 | ChIP, 5 µg (1:300) |
| Antibody | Taf1 antisera (rabbit polyclonal) | Joseph Reese and Song Tan, Pennsylvania State University | NA | ChIP, 2 µL (1:250) |
| Recombinant DNA reagent | pFA6a-FRB-kanMX6 | Euroscarf | P30578 | |
| Recombinant DNA reagent | pFA6a-2-FKB12-His3MX6 | Euroscarf | P30583 | |
| Recombinant DNA reagent | pSC11 | *Burke et al., 2000* | | |
| Commercial assay or kit | Magnetic Bead Isolation Kit | New England Biolabs | Cat# S1550S | |
| Commercial assay or kit | NEB Next Ultra RNA Library Prep Kit | New England Biolabs | Cat# E7530S | |
| Commercial assay or kit | NEBNext Ultra II Library Prep Kit | New England Biolabs | Cat# E7645S | |
| Commercial assay or kit | NEBNext Multiplex Oligos for Illumina | New England Biolabs | Cat# E7710 | |
| Commercial assay or kit | NEXTflex barcodes | BIOO Scientific | Cat# 514122 | |
| Chemical compound, drug | Rapamycin | LC Laboratories | Cat# R-5000 | |

*Continued on next page*

*Continued*

| Reagent type (species) or resource | Designation | Source or reference | Identifiers | Additional information |
|---|---|---|---|---|
| Software, algorithm | bwa | PMID: 10571001 | RRID: SCR_010910 | |
| Software, algorithm | SICER | PMID: 19505939 | RRID: SCR_010843 | |
| Software, algorithm | BioConductor | PMID: 15461798 | RRID: SCR_006442 | |
| Software, algorithm | Integrative Genomics Viewer | PMID: 21221095 | RRID: SCR_011793 | |
| Software, algorithm | tophat2 | PMID: 23618408 | RRID: SCR_013035 (Tophat) | |

## Yeast strains

Anchor away (AA) strains used in this study were constructed as described previously and contain the *tor1-1* mutation and deletion of *FPR1* (*Haruki et al., 2008*). To allow the AA technique, Taf1, TBP, and Rpb3 were chromosomally tagged with the FRB domain by lithium acetate transformation of PCR products amplified from pFA6a-FRB-kanMX6 into the AA parent strain (yFR1321, generous gift of F. Robert, Institut de Recherches Cliniques de Montréal) (*Hill et al., 1991*; *Longtine et al., 1998*; *Jeronimo and Robert, 2014*). Mediator subunits Med2-TAP, Med14-TAP, Med15-myc, Med18-myc, and Med17-myc were tagged by transformation of PCR products amplified from prior strains, as described (*Ansari et al., 2014*; *Paul et al., 2015*).

To generate double *kin28-taf1-AA* and *kin28-rpb3-AA* strains containing either Med15-myc or Med18-myc, the *kin28-AA* strain (KHW127, a generous gift of Kevin Struhl, Harvard Medical School) (*Wong et al., 2014*) was first switched from *MATα* to *MATa* using the pSC11 plasmid (*Burke et al., 2000*), and then mated with the *taf1-AA* and *rpb3-AA* strains, followed by sporulation and isolation of desired products. We were not able to obtain tetrads from *kin28-AA tbp-AA* diploids, and so this strain was constructed by introducing the FRB epitope tag directly into the *kin28-AA* parent strain using standard methods (*Longtine et al., 1998*). Similarly, *med2Δ med3Δ med15Δ* yeast were generated by crossing strains LS01H (*med2Δ*) and LS10 (*med3Δ med15Δ*) harboring a plasmid bearing the *URA3* gene, followed by sporulation, tetrad dissection, and identification of *med2Δ med3Δ med15Δ* haploids. A *kin28-AA* strain was constructed from one such segregant by integration of the *tor1-1* mutation, deletion of *FPR1*, and chromosomally tagging *RPL13A* with 2XFKBP and *KIN28* with FRB (*Haruki et al., 2008*).

Strain genotypes are provided in Table S1, and oligonucleotides used in strain construction are listed in Table S2. Loss of viability of anchor away strains in the presence of rapamycin is documented in *Figure 3—figure supplement 1*. For spot dilution assays, yeast cells were grown to an $OD_{600}$ 1.0 and serial dilutions were spotted on YPD medium and incubated at 30°C for 2–3 days, as indicated in the figure legends.

## RNA-seq

RNA was prepared from exponentially growing yeast cultures by the hot phenol method, using two independently derived *med2Δ med3Δ med15Δ* isolates and two biological replicate samples for wild type yeast (BY4741) (*Schmitt et al., 1990*). PolyA +RNA was prepared using a Magnetic Bead Isolation Kit (New England Biolabs), yielding from 1.3% to 9% recovery. Library preparation was performed using the NEB Next Ultra RNA Library Prep kit. Sequencing was performed on the Illumina NextSeq platform at the Wadsworth Center, New York State Department of Health.

## ChIP-seq

Whole cell extracts (WCE) of AA strains were prepared from 50 mL of culture grown in yeast peptone dextrose (YPD) at 30°C to doubling phase at an $OD_{600}$ of 0.8. Rapamycin (LC Laboratories, Woburn, MA) was then added one hour prior to crosslinking to a final concentration of 1 μg/mL

from a 1 mg/mL stock, stored in ethanol at −20 ℃ for not more than one month. (Concentration of rapamycin stock solutions was determined using $A_{267}$ = 42 and $A_{277}$ = 54 for a 1 mg/ml solution.) Immunoprecipitations were performed using the following antibodies: 5.0 µg Pol II unphosphory-lated CTD (8WG16, Biolegend, San Diego, CA), 1 µg Rpb3 (Neoclone), 2–5 µg c-Myc epitope (9E10; Roche, Sigma), 2.5 µg protein A (Sigma), 2.5–5 µg TBP (58C9, Abcam, or 5 µL serum, generous gift from A. Weil, Vanderbilt University), and 2.0 µL Taf1 (serum, generous gift from J. Reese and Song Tan, Penn State University). For spike-in experiments, whole cell extract was prepared from *S. pombe* strain FWP510 (*DeGennaro et al., 2013*) grown in Yeast Extract Supplemental (YES) medium and added to *S. cerevisiae* whole cell extract in a ratio of 1:4 to 1:5 prior to antibody incubation. At least two biological replicates (from independent cultures, on different days) were performed for all ChIP-seq experiments except as follows: Single ChIP-seq experiments were performed for *tbp-AA* yeast against Rpb1, with and without rapamycin; for ChIP against Med2-TAP and Med14-TAP in *tbp-AA* and *rpb3-AA* yeast; for Med17-myc in *kin28-tbp-AA* yeast; and for experiments targeting Rpb1 and Rpb3 (separately) in *rpb3-AA* yeast, with and without rapamycin. See Table S3 for details.

Library preparation for Illumina paired-end sequencing was performed with the NEBNext Ultra II library preparation kit (New England Biolabs) according to manufacturer's protocol and barcoded using NEXTflex barcodes (BIOO Scientific, Austin, TX) or NEBNext Multiplex Oligos for Illumina. In some experiments, a size selection step was performed on barcoded libraries by isolating fragment sizes between 200–500 bp on a 2% E-Gel EX agarose gel apparatus (ThermoFisher Scientific). Sequencing was performed on the Illumina NextSeq 500 platform at the Wadsworth Center, New York State Department of Health.

### ChIP-Seq and RNA-seq analysis

Unfiltered sequencing reads were aligned to the *S. cerevisiae* reference genome (Saccer3) using bwa (*Seoighe and Wolfe, 1999*). Up to one mismatch was allowed for each aligned read. Reads mapping to multiple sites were retained to allow evaluation of associations with non-unique sequences (*Seoighe and Wolfe, 1999*) and duplicate reads were retained. Binding peaks were identified using SICER (*Zang et al., 2009*) with the following parameters: effective genome size 0.97 (97% of the yeast genome is mappable,), window size 50 bp, and gap size 50 bp. Calculation of coverage, comparisons between different data sets, and identification of overlapping binding regions were preceded by library size normalization, and were performed with the 'chipseq' and 'GenomicRanges' packages in BioConductor (*Gentleman et al., 2004*). Control subtraction was carried out in the following way: coverage (exp)/N1 − coverage (control)/N2, in which 'exp' is the data set (in. bam format) to be examined, N1 is the library size of the experimental data ('exp'), and N2 is the library size of the control. For the spike-in experiments of *Figure 2*, normalization was performed as described (*Orlando et al., 2014*). Occupancy profiles were generated using the Integrative Genomics Viewer (*Robinson et al., 2011*). For metagene analysis, a list of 498 genes exhibiting Mediator ChIP signal upstream of promoters ('UAS genes') was downloaded from *Jeronimo et al. (2016)*; SAGA-dominated and TFIID-dominated genes were selected according to *Huisinga and Pugh (2004)*; and genes were sorted according to transcript levels using data from *Nagalakshmi et al. (2008)*.

RNA-seq reads were aligned to SacCer3 using tophat2 (*Kim et al., 2013*). Non-mRNA species were removed after mapping, and the total number of reads mapping to mRNA species were then calculated and mRNA coverage depth re-normalized. For metagene analysis, including heat maps, reads were normalized against input from *kin28*-AA yeast (KHW127) unless otherwise noted.

RNA-seq and ChIP-seq reads have been deposited in the NCBI Short Read Archive under project number PRJNA413080.

## Acknowledgements

This work was supported by grant MCB1516839 to RHM, and was also funded in part by the NIH Intramural Research Program at the National Library of Medicine (ZIZ and DL). We are grateful to Todd Benziger for help with strain construction. We thank Kevin Struhl, Francois Robert, and Fred Winston for yeast strains and reagents; Todd Gray, Tony Weil, Song Tan, and Joe Reese for generous provision of antibodies; and Joe Wade, Fred Winston, and Ameet Shetty for helpful discussions. We gratefully acknowledge help from the Wadsworth Center Applied Genomics Technology and Tissue Culture and Media Cores.

## Additional information

### Funding

| Funder | Grant reference number | Author |
| --- | --- | --- |
| National Science Foundation | MCB1516839 | Elisabeth R Knoll<br>Debasish Sarkar<br>Randall H Morse |
| National Institutes of Health | Intramural program | Z Iris Zhu<br>David Landsman |

The funders had no role in study design, data collection and interpretation, or the decision to submit the work for publication.

### Author contributions

Elisabeth R Knoll, Conceptualization, Formal analysis, Validation, Investigation, Visualization, Writing—original draft, Writing—review and editing; Z Iris Zhu, Data curation, Formal analysis, Visualization, Writing—original draft, Writing—review and editing; Debasish Sarkar, Validation, Investigation; David Landsman, Formal analysis, Supervision, Funding acquisition, Writing—review and editing; Randall H Morse, Conceptualization, Formal analysis, Supervision, Funding acquisition, Validation, Investigation, Writing—original draft, Writing—review and editing

### Author ORCIDs

Elisabeth R Knoll (iD) http://orcid.org/0000-0002-1083-6472
David Landsman (iD) http://orcid.org/0000-0002-9819-6675
Randall H Morse (iD) http://orcid.org/0000-0003-0000-8718

### Decision letter and Author response

Decision letter https://doi.org/10.7554/eLife.39633.028
Author response https://doi.org/10.7554/eLife.39633.029

## Additional files

### Supplementary files

• Supplementary file 1. Yeast strains used in this study.
DOI: https://doi.org/10.7554/eLife.39633.019

• Supplementary file 2. Primers used in this study.
DOI: https://doi.org/10.7554/eLife.39633.020

• Supplementary file 3. Summary of RNA-seq and ChIP-seq experiments.
DOI: https://doi.org/10.7554/eLife.39633.021

• Transparent reporting form
DOI: https://doi.org/10.7554/eLife.39633.022

### Data availability

Data from ChIP-seq and RNA-seq experiments have been deposited at the NCBI Short Read Archive under project number PRJNA413080.

The following dataset was generated:

| Author(s) | Year | Dataset title | Dataset URL | Database and Identifier |
| --- | --- | --- | --- | --- |
| Knoll ER, Zhu ZI, Sarkar D, Landsman D, Morse RH | 2018 | Data from ChIP-seq and RNA-seq experiments | https://www.ncbi.nlm.nih.gov/bioproject/PRJNA413080/ | NCBI BioProject, PRJNA413080 |

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
