## [Decision Letter]

[Editors’ note: a previous version of this study was rejected after peer review, but the authors submitted for reconsideration. The first decision letter after peer review is shown below.]

Thank you for submitting your work entitled "Role of the pre-initiation complex in Mediator recruitment and dynamics" for consideration by *eLife*. Your article has been reviewed by three peer reviewers and the evaluation has been overseen by a Reviewing Editor, Alan Hinnebusch, and a Senior Editor. The reviewers have opted to remain anonymous.

Our decision has been reached after consultation between the reviewers. Based on these discussions and the individual reviews below, we regret to inform you that your work will is not acceptable for publication in *eLife*.

The reviewers found intriguing the data obtained from anchor-away of TBP in the kin28-AA background, which seemed to reverse the normal transition of Mediator from UAS to promoter, whereas anchor-away of Rpb3 did not have this effect and resulted instead in loss of Mediator from the promoter. As you have concluded, these results suggest that the transition from UAS to promoter specifically requires TBP, such that Mediator appears stranded at the UAS in TBP's absence; whereas on depletion of Pol II, the TBP-mediated transfer presumably still occurs but Mediator is subsequently lost from the promoter and Mediator occupancy at both UAS and promoter is very low. Your results further suggested that Taf1 might be required along with Pol II to stabilize Mediator at the promoter; although it was felt that additional experiments are required to rule out the possibility that Taf1 is required instead for initial recruitment of Mediator to the UAS. There was also an appreciation of new findings regarding Mediator recruitment that are germane to the functional distinction between TFIID- versus SAGA-dependent genes.

Although the role of the PIC in stabilizing Mediator at promoters is strongly predicted by previous studies, uncovering distinct functions for TBP versus Pol II and Rpb3 in the UAS to promoter transition would represent novel findings. The difficulty is that other interpretations of the existing data are possible, and it was felt that significant additional experiments would be required to confirm the existing findings. One critically important question is whether the different outcomes elicted by depletion of TBP, Rpb3, or Taf1 merely reflect different quantitative degrees of PIC reduction rather than specific effects of depleting particular PIC components. It is important, therefore, to analyze the consequences of depleting each factor on the occupancies of the others, and ideally a fourth PIC component like TFIIB, to determine whether a coherent model can still be proposed in which TBP and Pol II, and Taf1 make distinct contributions to the UAS to promoter transition by Mediator.

There are also substantial differences in the data regarding the behavior of Med15 and Med8 on depletion of TBP versus Rpb3/Taf1, and there is a concern that the Med8 results are not accurately representing the behavior of Mediator head. Thus, additional head subunit(s) would have to be analyzed to confirm the Med8 findings and firmly establish whether or not different Mediator modules are affected differently by the three depletions. Finally, there were also several instances in which results were not presented comprehensively and, thus, doubts emerged about how representative the data being presented might be of the entire data set collected.

In view of the considerable amount of new experimental work that seems necessary, the extended time-frame this would likely entail, and the largely uncertain consequences of the new results for your final model, it was decided that the manuscript should be rejected rather than returned to you for revisions. The following list summarizes the most important criticisms, some of which were already mentioned above.

1) Repeat the Taf1 anchor away experiment in a WT KIN28 strain to determine whether Med15 occupancy at UASs is increased (as in Tbp3 depletion) or decreased, because if the latter was observed, this would implicate Taf1 in the initial recruitment of Mediator to promoters in addition to, or instead of, the transition of Mediator from UAS to PIC.

2) To confirm the differences seen between the Tail Med15 and Head Med18 occupancies on depletion of TBP in the kin28-AA strain in Figure 4C, examine the genome-wide behavior of additional Head subunits.

3) Compare the effects of depleting TBP, Rpb3, or Taf1 on PIC assembly to rule out the trivial possibility that the different outcomes on depletion of each factor represent different extents of reducing PIC assembly rather than indicating distinct functions in Mediator transfer or retention at the promoter. This would also address whether depletion of one factor by anchor away is simply more efficient than another.

4) Results for all four Mediator subunits, for all groups of genes, must be shown for a comprehensive analysis of the effects of depleting TBP vs. Rpb3 vs Taf1.

5) Provide a biological replicate for the results on Med2 in Figure 1A, as well as an input or untagged control.

6) Present time-courses of anchor away to justify the rapamycin treatments chosen for each protein.

Reviewer #1:

Summary of findings:

This study probes the role of PIC assembly, particularly the contributions of TBP, Taf1, and Pol II itself in the transition of Mediator from UAS to promoters in the course of the initiation of transcription in budding yeast. They show that a triple deletion of Mediator tail subunits is viable and allowed appreciable transcription with significant down-regulation of SAGA-dependent genes. In a kin28-AA strain, in which Mediator occupancy at promoters is enhanced, the triple tail deletion reduced Mediator Head occupancies at promoters and essentially abolished it at UAS elements. This finding largely confirms results/conclusions made previously from analysis of a double deletion of Med3/Med15 that the tail module strongly contributes to Mediator recruitment to the majority of genes, including those in which Mediator occupancy can be observed only on depletion of Kin28 and attendant shift of Mediator from UAS to promoter owing to loss of the CDK module, and its stabilization at the promoter owing to loss of Kin28 kinase activity.

They then show that depletion of TBP or Rpb3 actually increases WT Mediator tail subunit occupancy at UAS regions, while not affecting head subunit occupancy, for genes that show strong UAS occupancy of Mediator in WT cells, and also allows detection of Mediator at genes, e.g. RPGs, where it is not normally detected. This provides support for the model proposed by Struhl and Robert that, at genes for which Mediator association with UAS regions is observed, a single Mediator complex bridges the UAS and the promoter, with tail module subunits contacting the UAS and head and middle subunits engaging the promoter region through contacts with Pol II and GTFs; and suggests that interruption of the latter contacts by impairing PIC assembly (by depleting TBP or Rpb1) appears to strand Mediator at the UAS. These results also show that Mediator is actually present at genes (RPGs) where it was previously undetectable, as its transit from UAS to promoter is being blocked by impairing PIC assembly.

They go on to show that depleting TBP in cells where Kin28 is also depleted suppresses the effect of depleting Kin28 alone in shifting Mediator tail occupancy from UAS to PIC, providing evidence that this transition, which has been attributed to loss of the CDK module with attendant enhancement of Mediator interaction with the PIC at promoters, strongly depends on the presence of TBP in the PIC. However, depletion of Rpb3 in the kin28-AA background has a rather different effect-there is some shift from promoter to UAS, but the amount of tail signal (as well as head signal) overall is strongly reduced. This could be interpreted to indicate that Pol II is required for recruitment of Mediator per se, even to UAS elements; however, this possibility is disfavored by the fact that depletion of Rpb3 alone increases UAS occupancy of tail and scaffold subunits (Figure 2). Thus, they interpret the decrease in Mediator occupancy on Rbp3 depletion, seen only in the presence of Kin28 depletion, to result from a failure to retain Mediator at promoters following the shift from UAS to promoters that is evoked by depletion of the CDK module, thus implicating Pol II itself in stabilizing the PIC at the promoter. The shift of some mediator occupancy from promoter to UAS observed on depleting Rpb1 together with Kin28 is being interpreted (I believe) as an indirect result of reduced TBP occupancy on depletion of Rpb3 and attendant reduction in the TBP-dependent shift from UAS to promoter at certain genes; however the reduced TBP occupancy on depletion of Rpb3 should probably be confirmed experimentally if not already demonstrated previously.

Depleting Taf1 together with Kin28 has yet another consequence, of greatly reducing Mediator occupancy at the promoters, without evoking any shift back to the UAS. Thus, it seems that Taf1 could also be required for stabilizing Mediator at promoters and its depletion would have no effect on the shift from UAS to promoter and only on Mediator retention at promoters following the shift evoked by depletion of the CDK module. However, it seems possible that these results could be explained differently if Taf1 contributes to initial recruitment, which I think they are actually suggesting in panel B of the model in Figure 6. I believe that they should be asked to deplete Taf1 alone to determine whether Mediator occupancy at UASs is increased (as in Rpb3 depletion) or decreased, and if the latter is observed, this would implicate Taf1 in the initial recruitment of Mediator to promoters.

General critique:

While some of their results can be viewed as being largely confirmatory of previous findings from the Struhl and Robert labs, other findings increase our understanding of how Mediator is recruited to genes, and the transition of Mediator from the UAS to the promoter. Examination of the triple Mediator tail subunit deletion, in which the tail is completely eliminated provides stronger evidence that obtained previously with just two subunits deleted, that the tail module is important for Mediator recruitment at most active genes; but also reveal a tail-independent pathway that allows appreciable Mediator recruitment and transcription to continue, which they are able to argue involves Mediator recruitment by Pol II. Their findings that TBP is required for the transition of Mediator from UAS to promoter, even when the CDK module is depleted, whereas Pol II is important for retention of Mediator at promoters, and might contribute only indirectly to enhancing the UAS-promoter transition by enhancing TBP occupancy at the promoter, are significant. The role of Taf1 is clearly distinct from that of TBP, but it remains unclear whether Taf1 is enhancing Mediator recruitment to the UAS (like TBP) or stabilizing Mediator at the promoter after the UAS-promoter transition (like Pol II).

To resolve this last issue, the authors should conduct the additional experiment of depleting Taf1 alone to clarify its role in Mediator recruitment vs stabilization at promoters.

Reviewer #2:

The manuscript by Knoll and colleagues describes the effect of PIC components depletion on Mediator occupancy in budding yeast. They show that Mediator association with core promoters depends on the integrity of the PIC. Although this had never been formally demonstrated, this finding was expected based on the bulk of data available on Mediator-PolII and Mediator-GTFs interactions in vitro, including recent structural data. They also show that tampering with the PIC leads to increased detection of Mediator (although not all subunits tested) at UAS. This last finding was less expected and the fact that not all subunits behave the same is intriguing. The experiments are well executed and controlled. As is stands, however, the manuscript only represents a small increment of knowledge relative to current literature. Further exploring the effect of PIC depletion on Mediator-UAS interactions could potentially increase the impact of the work.

Specific comments:

First section entitled "Mediator is recruited to gene promoters in the absence of the tail module triad" is simply confirming data from Struhl and Robert labs. The only difference here is the use of a triple deletion of the triad (instead of a double deletion that leads to the loss of the third component). The data is very nice but constitutes a very small increment over published data.

The second section entitled "Mediator association at UAS regions is stabilized by loss of TBP or Pol II" shows that Mediator (Med2, Med15 and Med14) detection at UAS is increased upon disruption of the PIC by Rpb3 or TBP anchor-away. This observation is interesting and the authors interpret this data in the context of the "transition model" proposed by Struhl and Robert labs and suggest that interfering with the PIC leads to the stabilisation of Mediator at UAS. Unexpectedly, however, no change is observed with the Head subunit Med18. If the whole Mediator was stabilised at UAS as proposed by the authors, then all subunits should be affected. The Med18 data may suggest more complex interpretations. For instance, it may be that after its transient interactions with the PIC, Head/Middle dissociates, leaving the Tail stably associated with the UAS. In subsequent rounds, the Head/Middle would re-attach on the UAS-bound Tail and so on. This section of the manuscript has probably the most potential for novelty, but more experiments are needed. I would also suggest reorganizing Figure 2 (and Figure 4). The data for all subunits tested should be represented the same way. Why are Med18, Med2 and Med14 not fully represented as for Med15? Also, why are Med2 and Med14 mapped on SAGA and TFIID genes while the other subunits are mapped on "UAS genes".

The third and fourth sections "Mediator association with promoters depends on PIC components" and "Depletion of TBP, and to lesser extent Pol II, stabilizes Mediator occupancy at UAS regions" examine the effect of TBP, TAF1 and Rpb3 depletion on Mediator occupancy at promoters and UAS in kin28AA cells. I found the description of this data somewhat confusing and some of the conclusions not to be supported by the data. First, both sections really describe the same data so I do not see why separating them. It would be simpler to go through the data systematically and (as suggested for Figure 2) represent all datasets the same way (Figure 4). Currently, the third section describes TAF1 and Rpb3 and says nothing about TBP. It is not clear why. Second, it seems to me that all three depletions lead to the same phenotype, although with different severity. The TBP depletion is described as having fundamentally different effects on Mediator compared to Rpb3 and TAF1 depletions but one could argue that TBP depletion simply has a more pronounced effect on Mediator as a consequence of a more profound effect on the PIC. The authors should show the effect of their depletions on a few PIC components. One would expect TBP depletion to abolish the PIC altogether, whereas TAF1 and Rpb3 depletions may leave partial PIC behind. Alternatively, the differences in magnitude may be due to variable efficiencies (or timing) of depletion for TBP, Rpb3 and TAF1, a possibility not acknowledged. On a related note, the authors should show the depletion of their AA strains over time (easily done by ChIP-qPCR) in order to justify the time point used and relativize the effect observed for each of them. Third, I do not see what data allows the authors to conclude about timing of events as claimed in their discussion: "[…] TBP being most important for transit of Mediator from UAS to promoter and Rpb3 and Taf1 being more important for stabilizing occupancy at the promoter following transit".

Reviewer #3:

This paper describes a very nice set of genome-wide experiments that demarcate how and where Mediator is recruited to promoters. In one pathway the UAS/Activator complex recruits Mediator via the tail module to the UAS region, and then ultimately it sets down over the PIC. In the second pathway, which is UAS-independent and tail-independent, Mediator is directly recruited to the PIC, which most likely involves TFIID. The data and manuscript were presented clearly, and in general the conclusions were supported by the data. Overall the manuscript represents an important contribution to our understanding of Mediator, and will be valued by the scientific community.

Specific comments

1) It seems that single replicates were performed for some "main figure" experiments, including Figure 1A (Med2 TAP). Please confirm Figure 1A with a second biological replicate for Med2, since this is a main figure and drives the rationale for the rest of the manuscript. Also, is there an untagged/input control for Figure 1A? This would be helpful to get a sense of the recruitment of the tailless mutant over background.

2) There needs to be a correlation coefficient for Figure 1C to get a quantitative measure of difference in the gene expression in mutant relative to WT. "dispensable for most gene expression" would be more convincing with showing the numerical value for the graph.

3) Figure 2B, why isn't occupancy of Med 18 changed with AAR?

4) Anchor away was performed for 1 hr. Please provide an explanation for why this time frame, as opposed to less time like 20-30 min. Obviously, there is a trade-off between completeness of the sequestering and the potential for indirect effects to creep in. Was this condition chosen for a reason or was it arbitrary? Is there GFP localization experiment to verify the anchor away?

5) The results in Figure 4 are striking, and show that when Kin28 is depleted Mediator occupancy at TFIID promoters is highly dependent on Pol II and Taf1. I'm a bit puzzled as to why there needs to be a dependence on Kin28. Or is that a product of the experimental design, in that Taf1 or pol II AAR was not conducted in a WT KIN28 strain? Why are opposite effects seen for Med 15 and Med 18 in tbp+kin28 AAR?

[Editors’ note: what now follows is the decision letter after the authors submitted for further consideration.]

Thank you for submitting your article "Role of the pre-initiation complex in Mediator recruitment and dynamics" for consideration by *eLife*. Your article has been reviewed by three peer reviewers and the evaluation has been overseen by a Reviewing Editor, Alan Hinnebusch, and Kevin Struhl as the Senior Editor. The reviewers have opted to remain anonymous.

The reviewers have discussed the reviews with one another and the Reviewing Editor has drafted this decision to help you prepare a revised submission.

While two of the reviewers were satisfied with the additional work and revisions you made to the previous (rejected) version of the manuscript, and considered the results now to be worthy of publication after some minor revisions, reviewer #2 has raised serious concerns about the apparent lack of reproducibility between certain replicates, and a lack of clarity about whether results from two replicates have been used generally to draw key conclusions throughout the work. The recommended approach would be to draw conclusions after combining results from at least two highly correlated replicate datasets. Hence, it is possible that you would be obliged to conduct additional replicate experiments and re-evaluate your findings and conclusions after combining results from only highly correlated datasets. Reviewer #2 has also raised concerns that the AA tag on Taf1 is affecting its function in the absence of Rapamycin, and this complication has to be taken into account in the interpretation of results obtained in the presence of Rapamycin for this mutant, as the tag is setting a baseline of constitutive impairment against which the effect of the drug must be compared. Reviewer #3 has also raised some minor concerns that would also need to be addressed.

Reviewer #1:

The authors have done a considerable amount of additional work to address the major criticisms of the previous version. In particular, they carried out the experiment I had requested of depleting Taf1 in a strain with WT Kin28, and can now rule out a role for Taf1 in Mediator recruitment.

The work is significant for several reasons. Their findings indicating that TBP is required for transit of Mediator from UAS to promoter, whereas Taf1 of TFIID is dispensable for that function but appears to be as important as Pol II itself in retaining Mediator at promoters following the transfer from UAS are both important findings that could not have been predicted from previous work. It is also interesting that the importance of Taf1 and Pol II in retaining Mediator at promoters differs significantly in magnitude between TFIID and SAGA-dominated promoters. The additional work done to create the triple tail subunit deletion and quantify the effects of a completely tail-less mediator on recruitment of Mediator and Pol II occupancy is also valuable, as the effects are more severe than observed for the double tail subunit deletion. They also show unequivocally that the tail is not essential for Mediator recruitment, and that other interactions of Mediator with Pol II, TFIID, or TBP suffice for less efficient recruitment in the absence of the tail. On the other hand, the TBP anchor away experiments show that PIC formation is dispensable for Mediator recruitment in the presence of an intact Mediator tail, as Pol II recruitment is nearly abolished on TBP AA, with a shift in Mediator occupancy to UAS regions indicating that Mediator is being recruited but is stranded at the UAS in the absence of TBP's function in Mediator transit. These latter experiments also reveal that Mediator is recruited to the UASs of many genes where it cannot normally be detected.

I have no additional criticisms.

Reviewer #2:

The manuscript by Knoll et al. describes how yeast Mediator occupancy at UAS and promoters is affected by PIC components. Using combinations of anchor-away experiments coupled to ChIP-seq, they propose that TBP is required for the transit of Mediator from UAS to promoters while Pol II and Taf1 are required for stabilising Mediator at promoters. They also provide data showing that the Tail triad is not required for the association of Mediator with promoters.

This work is building on prior work from the Struhl and Robert lab and several aspects are largely confirmatory. The most novel aspect concerns how different PIC components contribute to the association of Mediator with promoters. Unfortunately, the data appears to be of variable quality and the conclusions are based on interpreting cherry-picked replicates while ignoring others that would have led to very different conclusions.

My understanding is that some experiments were done as only one replicate while others were done in duplicates. The main problem is that the analysis appears to have been done on one of the replicates (which is shown in the main Figures) while the second replicate is (at least in some occasions and it is not clear whether this is systematic) shown as a figure supplement and not used to build the story. Normally, experiments are done at least in duplicate and some correlation metrics are used to insure they are alike. Then replicates are combined and analysed further. Here, not only the correlation between replicates has not been assessed but one is used and the other relayed to a sup figure. Yet, at least in some cases, the replicates are so different that considering the alternate would have led to a completely different story. Here is a striking example: Figure 3A shows that Med15 occupancy is massively increased upon anchor-away of TBP. As the authors emphasised, this is radically different from Med18, the occupancy of which is not affected by the anchoring of TBP (as shown in Figure 3B). Yet, the second replicate of these experiments, shown in Figure 3—figure supplement 3, shows no effect of TBP anchoring on Med15 (rather that a massive increase) and a decrease for Med18 (rather than no change). Similar inconsistencies are also present for the rpb3 and taf1 anchoring on the same sets of Figures. The massive increase of Med15 occupancy upon TBP anchoring being the founding data for the story, I am finding it difficult to consider the rest of the story. In addition to Figure 3, the main results of this paper are those in Figure 5. In this case, I was not able to figure out whether the data presented in Figure 5 was for 1 rep out 1, 1 rep our 2 (in which case rep2 is nowhere to be found) or a combination of 2 reps. Given the variability between reps observed in Figure 3, and not knowing about the replicate status of Figure 5, I am not confident in interpreting data from Figure 5.

Another major concern with the manuscript has to do with the Taf1 anchor-away data. As acknowledged by the authors, the taf1-FRB strain appears to show a phenotype even in the absence of rapamycin. This means that the tagging affected the function of the protein. Because rapamycin did not lead to any addition effect, I do not think that the data can be interpreted as a consequence of conditional depletion. Unless the authors find a way to conditionally affect the function of Taf1 (perhaps auxin degron or tagging FRB in N-terminal would work better), I believe the Taf1 data has to be removed.

In sum, I recommend the authors go back to the basics and make sure their experiments are reproducible. For that, they need to show that their replicates are highly correlated. At current state, I suspect that most of the effects observed are batch effects rather than biological effects. For instance, simply inspecting the red tracks in Figure 3—figure supplement 5A clearly shows that rep2 did not work nearly as well as rep1. Not seeing the labels, one could not even tell how to pair them. Also, looking at the blue tracks from the same Figures suggests that the samples cluster by batches rather than by conditions.

Reviewer #3:

In this revised manuscript, the authors show that mediator recruitment to upstream activated regions (UAS) via transcription activators may be dependent or independent of the tail module. Importantly their triple KO when compared to WT has reduced occupancy of mediator complex (via Med17 ChIP) and PolII (via Rbp1 ChIP) at most genes, but occupancy of both is present in the absence of the Med2, Med3, and Med15. This reinforces the role of the Tail in binding to UAS and that the UAS enhances transcription, but importantly highlighting that even without Med2, there is minor mediator occupancy at UAS under conditions of basal transcription. This point is not merely confirmatory but key to understanding the overall role of mediator in basal and stress-induced transcription not only in yeast, but even in mammals akin to the role of enhancers enhancing transcription through the tail when needed. They show increased occupancy of Med15 (tail) with depletion of Tbp or Rpb3, but this is not noted for Med18 (head) or Med14 (backbone). This may suggest that PIC disruption may lead to Mediator not moving from the UAS region to the promoters (being stalled). Further, they show that Med15 and Med18 occupancy is decreased more at TFIID-dominated genes than SAGA-dominated genes upon Taf1 and Rpb3 depletion in the context of Kin28 anchor away. TBP depletion causes an upstream shift of mediator occupancy at SAGA-dominated genes (corresponding to UAS) and usually a loss of mediator occupancy at TFIID-dominated genes. Rpb3 loss leads to decreased occupancy, particularly at TFIID-dominated genes and an upstream shift of Med15 occupancy at SAGA-dominated genes with concomitant Kin28 depletion.

This paper clearly shows key mediator occupancy differences at SAGA- and TFIID-dominated genes in connection with PIC stabilization that is in line with many other papers. Importantly, they connect the mediator tail module to SAGA-dominated genes via their triple Tail KO. They demonstrate that the movement of the mediator from UAS to promoter is dependent on TBP and that the mediator is stabilized at promotors especially in TFIID-dominated genes by Taf1. While the model of dependent and independent recruitment of mediator by its tail module is plausible, the differentiation of mediator dynamics at SAGA- vs. TFIID-dominated genes is a very exciting aspect of this paper and nicely highlights mediator movement after initial recruitment in the context of PIC components.

Multiple aspects of this paper connect other previously isolated papers such as SAGA & TFIID-dominated genes, UAS and TSS distance in relationship to mediator, and PIC-component interactions with mediator modules; in so doing, they put minor aspects of key mechanisms at the UAS-promoter looping into context of each other to present a clear and more encompassing model. We greatly appreciate them starting to perturb the assumed complex relationship between PIC components and mediator and think their data will enable others to further explore these relationships. Moreover, the authors have sufficiently addressed our concerns from the previous review. We find this paper not to be merely confirmatory but addressing key issues of mediator biology.

[Editors' note: further revisions were requested prior to acceptance, as described below.]

Thank you for resubmitting your work entitled "Role of the pre-initiation complex in Mediator recruitment and dynamics" for further consideration at *eLife*. Your revised article has been favorably evaluated by Kevin Struhl (Senior Editor), a Reviewing Editor, and one reviewer.

The manuscript has been improved with the addition of new datasets, but there are several remaining issues raised by reviewer #2 that need to be addressed before acceptance. Please make appropriate revisions of text in response to all of the criticisms/comments in the review below and provide a point-by-point explanation of your responses along with the revised manuscript.

Reviewer #2:

The revised manuscript is improved, notably by the addition of datasets completing some that were done only once and by the replacement of non-consistent replicates. However, there are a few issues that are still of concern and should be addressed.

First, some of the anchor-away strains behave strangely. While this is unlikely to drive erroneous conclusions, it should at least be mentioned. For instance, Kin28-TBP-AA Med15-Myc is very sick according to Figure 3—figure supplement 1. More worrisome, tagging various PIC components with FRB greatly affects Mediator ChIP in the absence of rapamycin. This is evident by looking at several figures, notably in Figure 3 (pay attention to the y-axis for all the no rapa traces (black) or look at the intensities of the no rapa heatmaps). This is also directly shown in Fig 3—figure supplement 2B. The authors use this figure supplement to claim that Taf1-AA leads to increased Mediator ChIP signal in absence of rapamycin but they fail to acknowledge that this is also the case for TBP-AA (the average signal is 2 to 3 fold up compared to non-tag). It is not clear whether these differences are batch differences (due to technical variation between experiments performed on different days) or biological differences (caused by partial loss of function of tagged proteins).

Second, the Med18 ChIP data displayed in Figure 3 and showing that Med18 does not change in TBP-AAR or Rpb3-AAR does not fit the author's model very well. In subsection “Mediator association at UAS regions is stabilized by loss of TBP or Pol II” the authors say "The increase in ChIP signal observed for Med15, but not for Med18, would be consistent with the most immediate contacts with UAS-bound activators occurring through tail module subunits (such as Med15), while ChIP signal for Med18 at UAS regions would depend on protein-protein cross-links, thus lowering the efficiency of ChIP for this subunit". While I agree that this explains the lower ChIP efficiency of Med18 at UAS, it does not explain why, unlike other subunits tested, Med18 signal does not increase at UAS in TBP-AAR and Rpb3-AAR. The observed Med18 ChIP signal is clearly over the UAS so if Mediator is stuck there, this Med18 signal should increase together with other subunits. The fact that it does not (assuming the data is real), suggests a more complicated model. This result should at least be described more critically.

Third, the conclusion that TBP is required for the transition while Rpb3 and Taf1 are stabilising the PIC-associated Mediator is in my opinion a hazardous extrapolation. As explained by the authors in subsection “TBP is required for transit of Mediator from UAS to promoter”, depletion of PIC components has impact on other PIC components. This makes the attribution of specific roles to specific PIC components in Mediator dynamics difficult. Hence, with these limitations in mind, it seems like an over interpretation to say "TBP being critical for transit of Mediator from UAS to promoter, while Pol II and Taf1 stabilize Mediator association at proximal promoters". A more conservative interpretation would be that the PIC is necessary to allow Mediator transiting to the promoter (as shown when PIC is severely compromised; TBP-AA) and that partially formed PICs (without trying to attribute specific roles to subunits or proteins), while allowing the transit, lead to unstable Mediator-PIC association.

These last two points had been raised in previous review rounds but are still not clearly dealt with in the most recent version of the manuscript.

Finally, it is not clear why the authors were expecting a decrease of Mediator at UAS upon depletion of TBP, Rpb3, or Taf1. Isn't an increase what the "dynamic model" proposed by the Struhl and Robert labs predicted?

---

## [Author Response]

[Editors’ note: the author responses to the first round of peer review follow.]

[…] In view of the considerable amount of new experimental work that seems necessary, the extended time-frame this would likely entail, and the largely uncertain consequences of the new results for your final model, it was decided that the manuscript should be rejected rather than returned to you for revisions. The following list summarizes the most important criticisms, some of which were already mentioned above.1) Repeat the Taf1 anchor away experiment in a WT KIN28 strain to determine whether Med15 occupancy at UASs is increased (as in Tbp3 depletion) or decreased, because if the latter was observed, this would implicate Taf1 in the initial recruitment of Mediator to promoters in addition to, or instead of, the transition of Mediator from UAS to PIC.

We have included Taf1 anchor away experiments as requested; these are included in the revised Figure 3 (formerly Figure 2). These experiments clearly show that there is no decrease in Mediator occupancy at UAS regions upon depletion of Taf1.

2) To confirm the differences seen between the Tail Med15 and Head Med18 occupancies on depletion of TBP in the kin28-AA strain in Figure 4C, examine the genome-wide behavior of additional Head subunits.

We have included new experiments (Figure 5—figure supplement 4) examining the occupancy of two additional Mediator subunits, Med17 from the head module and Med2 from the tail module, after simultaneous depletion of Kin28 and TBP. The results corroborate those obtained with Med15 and Med18.

3) Compare the effects of depleting TBP, Rpb3, or Taf1 on PIC assembly to rule out the trivial possibility that the different outcomes on depletion of each factor represent different extents of reducing PIC assembly rather than indicating distinct functions in Mediator transfer or retention at the promoter. This would also address whether depletion of one factor by anchor away is simply more efficient than another.

We have included data on the effect of depleting PIC components on occupancy of other PIC subunits, and also cited relevant literature on this point (we had cited a bioRxiv paper on this previously, which has now been published in Genes & Development). Specifically, we document depletion of TBP, Rpb3, and Taf1 (which is less complete) in the corresponding anchor away strains in Figure 3—figure supplement 4 and 5. We also show the effects on occupancy of Pol II following depletion of TBP (Figure 3—figure supplement 4D), TBP and Kin28 (Figure 7A), and Taf1 and Kin28 (Figure 7A); on occupancy of Taf1 after depletion of TBP or TBP and Kin28 (Figure 7B); on occupancy of Taf1 after depletion of Rpb3 (Figure 7C); and on occupancy of TBP after depletion of Rpb3, and occupancy of Rpb1 after depletion of TBP (Figure 3—figure supplement 4). The implications of these results on interpreting Mediator occupancy changes after PIC component depletion are considered towards the end of the Results section.

4) Results for all four Mediator subunits, for all groups of genes, must be shown for a comprehensive analysis of the effects of depleting TBP vs. Rpb3 vs Taf1.

The results for the effects on occupancy of four Mediator subunits following depletion of TBP or Rpb3 are shown in Figure 3 (formerly Figure 2) in the revised manuscript, and results for two subunits are shown for Taf1 (all that we have done), as little change was seen for Taf1, and the major point for that PIC component is that recruitment of Mediator is not decreased upon its depletion. These results are now presented in the same way (at “UAS genes”) in all cases.

5) Provide a biological replicate for the results on Med2 in Figure 1A, as well as an input or untagged control.

We conducted a biological replicate for the results in Figure 1A of the original submission, and discovered that the signal we observed for Med2 occupancy was not significantly different to that seen with a control, untagged strain. We therefore have removed this part of Figure 1 from the manuscript.

6) Present time-courses of anchor away to justify the rapamycin treatments chosen for each protein.

We have presented a time course for anchor away of TBP, and also provide justification of our choice of one hour depletion by comparison with other published work using this method to deplete components of the transcription machinery.

In addition, we have included new data showing that Pol II occupancy is reduced genome-wide to about 25% of wild type levels in the *med2∆ med3∆ med15∆* mutant, demonstrating that the tail module plays an important role in Pol II occupancy at essentially all genes.

Reviewer #1:[…] While some of their results can be viewed as being largely confirmatory of previous findings from the Struhl and Robert labs, other findings increase our understanding of how Mediator is recruited to genes, and the transition of Mediator from the UAS to the promoter. Examination of the triple Mediator tail subunit deletion, in which the tail is completely eliminated provides stronger evidence that obtained previously with just two subunits deleted, that the tail module is important for Mediator recruitment at most active genes; but also reveal a tail-independent pathway that allows appreciable Mediator recruitment and transcription to continue, which they are able to argue involves Mediator recruitment by Pol II. Their findings that TBP is required for the transition of Mediator from UAS to promoter, even when the CDK module is depleted, whereas Pol II is important for retention of Mediator at promoters, and might contribute only indirectly to enhancing the UAS-promoter transition by enhancing TBP occupancy at the promoter, are signficant. The role of Taf1 is clearly distinct from that of TBP, but it remains unclear whether Taf1 is enhancing Mediator recruitment to the UAS (like TBP) or stabilizing Mediator at the promoter after the UAS-promoter transition (like Pol II).To resolve this last issue, the authors should conduct the additional experiment of depleting Taf1 alone to clarify its role in Mediator recruitment vs stabilization at promoters.

We thank the reviewer for a thoughtful and thorough critique, and are pleased that s/he finds our results significant. The reviewer states that the experiments using the *med2∆ med3∆* med15∆ are “largely confirmatory” of previous work from the Struhl and Robert labs, but also states that this result “provides stronger evidence than obtained previously”. We agree with both of these comments, mostly; although the effect of the triple deletion is not unexpected, the results establish definitively that the tail module triad is non-essential, contributes considerably to Mediator recruitment at most active genes, and demonstrates that alternative modes of Mediator recruitment and gene activation, independent of the tail module triad, exist. Prior to this work, the possibility existed that the tail module triad was essential, and that residual function of Med2 was sufficient to allow viability in the double mutant; this is now conclusively ruled out. We have also added a new experiment showing that Pol II occupancy is reduced to ~25% of wild type levels in the *med2∆ med3∆ med15∆* mutant across the genome.

With regard to the last point, that it is unclear whether Taf1 is important for Mediator recruitment to the UAS or for stabilizing Mediator at the promoter after transit from the UAS, we have included in the revised manuscript experiments that provide evidence that Mediator continues to occupy UAS regions after Taf1 depletion, in the presence of active Kin28. These data support the idea that Taf1 is important for stabilization of Mediator at promoters after transit from the UAS.

Reviewer #2:[…] First section entitled "Mediator is recruited to gene promoters in the absence of the tail module triad" is simply confirming data from Struhl and Robert labs. The only difference here is the use of a triple deletion of the triad (instead of a double deletion that leads to the loss of the third component). The data is very nice but constitutes a very small increment over published data.

We are pleased that the reviewer considers our data to be solid. We disagree that the first section, regarding the knockout of the complete tail module triad, is “simply confirming data from Struhl and Robert labs.” Our finding that Mediator occupancy at UAS regions in *med2∆ med3∆ med15∆* yeast is nearly completely eliminated can fairly be regarded as confirmatory, as a similar result was reported for the double mutant (*med3∆ med15∆)* and the clear expectation would be that loss of Mediator occupancy at UAS regions would be seen in the triple mutant also. However, it could not be predicted with confidence that the triple mutant would be viable, nor what the effect would be on Mediator occupancy at promoters. Our results provide the first definitive evidence that the Mediator tail module triad is not essential, and that Mediator can be recruited to gene promoters in its absence. We also show in the revised manuscript that association of both Mediator and Pol II is substantially decreased in the triple mutant, considerably more than the less than two-‐fold effect reported previously (Jeronimo et al., 2016), showing that the tail module triad is broadly important for recruitment of these components.

We agree with the reviewer that based on the in vitro results of Carey and co-workers, it seemed likely that Mediator association with gene promoters would depend on PIC integrity. But in vitro results do not always accurately predict in vivo interactions, and this prediction had not previously been tested experimentally.

The second section entitled "Mediator association at UAS regions is stabilized by loss of TBP or Pol II" shows that Mediator (Med2, Med15 and Med14) detection at UAS is increased upon disruption of the PIC by Rpb3 or TBP anchor-away. This observation is interesting and the authors interpret this data in the context of the "transition model" proposed by Struhl and Robert labs and suggest that interfering with the PIC leads to the stabilisation of Mediator at UAS. Unexpectedly, however, no change is observed with the Head subunit Med18. If the whole Mediator was stabilised at UAS as proposed by the authors, then all subunits should be affected. The Med18 data may suggest more complex interpretations. For instance, it may be that after its transient interactions with the PIC, Head/Middle dissociates, leaving the Tail stably associated with the UAS. In subsequent rounds, the Head/Middle would re-attach on the UAS-bound Tail and so on. This section of the manuscript has probably the most potential for novelty, but more experiments are needed. I would also suggest reorganizing Figure 2 (and Figure 4). The data for all subunits tested should be represented the same way. Why are Med18, Med2 and Med14 not fully represented as for Med15? Also, why are Med2 and Med14 mapped on SAGA and TFIID genes while the other subunits are mapped on "UAS genes".

We have remade Figure 2 (Figure 3 in the revised manuscript) so that it shows heat maps and line graphs at “UAS genes” for all four Mediator subunits. Because depletion of Taf1 showed little effect on Med15 and Med18, we did not examine the effect of Taf1 depletion on Med2 and Med14. As to the possibility that the tail module might interact with UASs in the absence of the rest of Mediator, we do mention this in the text but do not favor this possibility. First, the increased occupancy seen for Med14 upon depletion of Rpb3 or TBP (Figure 3D) is consistent with results for Med15 and Med2, and suggests that Mediator as a whole shows increased occupancy under these conditions. We believe this interpretation is most consistent with results from the Struhl and Robert labs, and in particular with the model shown in Figure 7 of Jeronimo et al., 2016. We show in Figure 3—figure supplement 5B that, as in the Jeronimo et al. paper, we observe ChIP signal at some genes after Kin28 depletion at both UAS and promoter regions for Med15, but only at promoter regions for Med17 and Med18. When considered together with the re-ChIP experiments in Petrenko et al., 2016, Figure 1, these results seem most consistent with Med18 exhibiting weaker ChIP signal at UAS regions than does Med15.

The third and fourth sections "Mediator association with promoters depends on PIC components" and "Depletion of TBP, and to lesser extent Pol II, stabilizes Mediator occupancy at UAS regions" examine the effect of TBP, TAF1 and Rpb3 depletion on Mediator occupancy at promoters and UAS in kin28AA cells. I found the description of this data somewhat confusing and some of the conclusions not to be supported by the data. First, both sections really describe the same data so I do not see why separating them. It would be simpler to go through the data systematically and (as suggested for Figure 2) represent all datasets the same way (Figure 4). Currently, the third section describes TAF1 and Rpb3 and says nothing about TBP. It is not clear why. Second, it seems to me that all three depletions lead to the same phenotype, although with different severity. The TBP depletion is described as having fundamentally different effects on Mediator compared to Rpb3 and TAF1 depletions but one could argue that TBP depletion simply has a more pronounced effect on Mediator as a consequence of a more profound effect on the PIC. The authors should show the effect of their depletions on a few PIC components. One would expect TBP depletion to abolish the PIC altogether, whereas TAF1 and Rpb3 depletions may leave partial PIC behind. Alternatively, the differences in magnitude may be due to variable efficiencies (or timing) of depletion for TBP, Rpb3 and TAF1, a possibility not acknowledged. On a related note, the authors should show the depletion of their AA strains over time (easily done by ChIP-qPCR) in order to justify the time point used and relativize the effect observed for each of them. Third, I do not see what data allows the authors to conclude about timing of events as claimed in their discussion: "[…] TBP being most important for transit of Mediator from UAS to promoter and Rpb3 and Taf1 being more important for stabilizing occupancy at the promoter following transit".

We agree that depletion of TBP is likely to have a more profound effect on the PIC than depletion of Rpb3 or Taf1, but we are puzzled by the statement that all three depletions lead to the same phenotype, but with different severity. The data in Figures 4C and D (Figure 5 in the revised manuscript) show that both Med15 and Med18 peaks shift upstream upon depletion of TBP at SAGA-dominated genes and genes having upstream Mediator occupancy in wild type yeast (“UAS genes”); Med15 can be seen to shift upstream at TFIID-dominated genes as well. Depletion of Rpb3 or Taf1 leads to a partial upstream shift or no upstream shift, respectively, but a strong decrease in signal. If TBP depletion simply had a more pronounced effect than depletion of Taf1, for instance, it should result in a stronger decrease and no upstream shift, or Taf1 should give rise to a partial upstream shift. These results indicate that TBP is necessary for Mediator occupancy to shift from UAS to promoter, but cannot determine whether it is sufficient, as Rpb3 occupancy is abolished by loss of TBP (Figure 3—figure supplement 4D and Figure 7A). In contrast, Mediator occupancy does not shift upstream upon depletion of Taf1, and only partially upon depletion of Rpb3, suggesting that these PIC components are not necessary for Mediator to transit from UAS to promoter region. Hence their behavior is qualitatively different. I believe the other two reviewers agree with this point.

We have included new data and citations to show the effect of individual PIC components on each other both with and without Kin28 depletion (Figure 3—figure supplement 4 and Figure 7). As the reviewer suggests, depletion of TBP eliminates Pol II association, while depletion of Rpb3 only causes partial loss of TBP, consistent also with recently published work (Joo et al., 2017). Perhaps surprisingly, Taf1 occupancy was only modestly reduced upon depletion of TBP or Rpb3 (Figure 7). We have added text to the Results that considers the results seen on Mediator occupancy in light of cooperative assembly of the PIC.

With regard to our rationale for using a one hour treatment with rapamycin, we chose this as a reasonable balance that would allow near complete depletion but not allow much time for indirect effects, as reviewer #3 suggests. Our ChIP data for anchoring away TBP indicated that depletion was essentially complete after 30 minutes, and we have now included this experiment as supplemental data. Other studies using anchor away to deplete components of the transcription machinery have used similar times of rapamycin treatment (30 min for Taf proteins (Warfield et al., 2017), 1 hr for Spt7 (Baptista et al., 2017) and for Mediator subunits, TBP, and Rpb1 (Wong and Struhl, 2014; Petrenko, 2016, 2017); 90 min for Mediator subunits (Jeronimo et al. 2016); 120 min for Mediator subunits (Anandhakumar et al., 2016)). Interestingly, the latter study found only about 80% depletion of Mediator subunits after 1 hr of rapamycin treatment, whereas the Struhl laboratory found depletion to background levels in 1 hr (Wong and Struhl, 2014), although the two studies used different criteria to monitor depletion. It is possible that the GFP tags also employed in the former study affected kinetics of depletion, but this has not been rigorously investigated.

Finally, with regard to organization, we prefer to keep the two sections mentioned separate. The first reports the evidence that Rpb3 and Taf1 are required for stable association of Mediator at promoters, while the second reports on the shift to the UAS region seen upon depletion of TBP. These distinct outcomes are best emphasized, in our opinion, by being discussed in separate sections.

Reviewer #3:[…] 1) It seems that single replicates were performed for some "main figure" experiments, including Figure 1A (Med2 TAP). Please confirm Figure 1A with a second biological replicate for Med2, since this is a main figure and drives the rationale for the rest of the manuscript. Also, is there an untagged/input control for Figure 1A? This would be helpful to get a sense of the recruitment of the tailless mutant over background.

We repeated the experiment of Figure 1A and included an untagged control. The signal seen in this experiment for Med2 at UAS regions was small (as in the original figure), but importantly not different from that seen with the untagged control. We have therefore not included this figure in the revised manuscript. However, we do not (and did not) consider this to be essential as a rationale for examining the complete deletion of the tail module triad. Prior to the work reported here, it remained possible that Med2 could contribute to Mediator recruitment to UAS regions and/or promoters in med3∆ med15∆ yeast; we have now formally excluded this possibility, while also defining rigorously the extent to which the tail module triad contributes to Mediator and Pol II recruitment genome-wide.

With regard to other experiments, in all cases where exact replicate experiments were not performed, we have performed corroborating experiments by doing ChIP against multiple Mediator subunits under the same conditions (e.g. Figure 3), or have relied on published results that corroborate our experiments (e.g. decreased occupancy of TBP upon depletion of Pol II). We have also added new experiments in which occupancy of two additional Mediator subunits, Med2 and Med17, is examined after depletion of Kin28 and TBP. The results are consistent with those observed for Med15 and Med18, with the tail module subunit, Med2, exhibiting greater signal at the UAS following depletion, while the head module subunit, Med17, shows strongly decreased signal relative to depletion of Kin28 alone, but very little upstream signal.

2) There needs to be a correlation coefficient for Figure 1C to get a quantitative measure of difference in the gene expression in mutant relative to WT. "dispensable for most gene expression" would be more convincing with showing the numerical value for the graph.

We have added correlation coefficients to the graphs in question and calculated Fisher’s z-score for the difference between correlation coefficients of the two graphs, yielding a p-value < 10^-10^. However, we have revised our conclusion regarding the role of the tail module in gene expression based on the new data showing that Pol II association is substantially (about 4‐fold) reduced in the tail module deletion mutant: evidently the tail module is broadly important for Pol II recruitment, but the effect of this reduced recruitment is likely offset by altered RNA decay, as has been observed for other perturbations in PIC components.

3) Figure 2B, why isn't occupancy of Med 18 changed with AAR?

We do not have a definitive answer for this question, but speculate that it may reflect Med18 being less proximate to UAS regions and correspondingly exhibiting a weaker ChIP signal there, as proposed by Jeronimo et al., 2016; see also our response to reviewer #2. Alternatively, Mediator may exist in a configuration that decreases accessibility to Med18 under the conditions of TBP depletion. Reviewer #2 suggests it could reflect loss of the Mediator head module (or perhaps only of specific subunits), but we do not favor this explanation at present as there is little evidence to support it.

4) Anchor away was performed for 1 hr. Please provide an explanation for why this time frame, as opposed to less time like 20-30 min. Obviously, there is a trade-off between completeness of the sequestering and the potential for indirect effects to creep in. Was this condition chosen for a reason or was it arbitrary? Is there GFP localization experiment to verify the anchor away?

We chose 1 hr as a reasonable balance that would allow near complete depletion but not allow much time for indirect effects, as the reviewer suggests. Our ChIP data for anchoring away TBP indicated that depletion was essentially complete after 30 minutes, and we have now included this experiment as supplemental data. Other studies using anchor away to deplete components of the transcription machinery have used similar times of rapamycin treatment (30 min for Taf proteins (Warfield et al., 2017), 1 hr for Spt7 (Baptista et al., 2017) and for Mediator subunits, TBP, and Rpb1 (Wong and Struhl, 2014;Petrenko 2016, 2017); 90 min for Mediator subunits (Jeronimo et al., 2016); 120 min for Mediator subunits (Anandhakumar et al., 2016)). Interestingly, the latter study found only about 80% depletion of Mediator subunits after 1 hr of rapamycin treatment, whereas the Struhl laboratory found depletion to background levels in 1 hr (Wong and Struhl, 2014). It is possible that the GFP tags also employed in the former study affected kinetics of depletion, but this has not been rigorously investigated.

5) The results in Figure 4 are striking, and show that when Kin28 is depleted Mediator occupancy at TFIID promoters is highly dependent on Pol II and Taf1. I'm a bit puzzled as to why there needs to be a dependence on Kin28. Or is that a product of the experimental design, in that Taf1 or pol II AAR was not conducted in a WT KIN28 strain? Why are opposite effects seen for Med 15 and Med 18 in tbp+kin28 AAR?

Depletion of kin28 in the experiment of Figure 4 was used to allow detection of Mediator at promoters of active genes, as shown by the Robert and Struhl labs (Jeronimo and Robert, 2014; Wong and Struhl, 2014). We have edited the text to clarify this point. We did include an experiment in which Pol II was depleted (“pol II AAR”), shown in Figure 2 (now Figure 3), and have added data for Taf1 depletion in the presence of wild type KIN28 to Figure 3.

I believe that what the reviewer is referring to as “opposite effects seen for Med15 and Med18 in tpb + kin28 AAR” is the apparent increased signal at SAGA-dominated genes for Med15 and decrease for Med18. We believe that this reflects Mediator being stranded at the UAS region after TBP depletion; because the contact with Med15 is much closer, ChIP efficiency is high and the signal is strong, whereas the Med18 subunit is more proximate to the promoter in kin28-‐AAR conditions than to the UAS in kin28-tbp-AAR conditions, resulting in decreased ChIP efficiency and lower signal. This interpretation is consistent with the model proposed by the Struhl and Robert labs, and specifically with the detailed structural considerations discussed in Jeronimo et al., 2016. We have added text to clarify this interpretation. See also Figure 3—figure supplement 5B, and our response to reviewer #2.

[Editors' note: the author responses to the re-review follow.]

Reviewer #2:The manuscript by Knoll et al. describes how yeast Mediator occupancy at UAS and promoters is affected by PIC components. Using combinations of anchor-away experiments coupled to ChIP-seq, they propose that TBP is required for the transit of Mediator from UAS to promoters while Pol II and Taf1 are required for stabilising Mediator at promoters. They also provide data showing that the Tail triad is not required for the association of Mediator with promoters.This work is building on prior work from the Struhl and Robert lab and several aspects are largely confirmatory. The most novel aspect concerns how different PIC components contribute to the association of Mediator with promoters. Unfortunately, the data appears to be of variable quality and the conclusions are based on interpreting cherry-picked replicates while ignoring others that would have led to very different conclusions.

We felt this reviewer was unduly harsh in accusing us of “interpreting cherry-picked replicates while ignoring others that would have led to very different conclusions”. This statement is based on the replicate data shown in Figure 3—figure supplement 3 in the critiqued manuscript, which included data (high signal for both Med15 and Med18 in the TBP anchor away strain in the absence of rapamycin) that was inconsistent with data in the main figure. In the text of the previously submitted version we argued that the preponderance of our results were most consistent with the interpretation we provided; it is not in our opinion accurate to state that we ignored the inconsistent results. Nonetheless the reviewer’s concern regarding reproducibility is valid and was a concern to us as well; that is why we chose to include replicate data for this experiment as a supplementary figure. We have now repeated this experiment—ChIP-seq of Med15 and of Med18 in the tbp-AA yeast strain in the absence and presence of rapamycin—and the results are consistent with those we had presented in Figure 3, including results for ChIP against Med2 and Med14 (see Author response image 1). We have revised Figure 3 by presenting data for Med15 and Med18 ChIP averaged from two independent biological replicate experiments (except for the Med18 ChIP in the taf1-AA strain, for which we have only one replicate).

**Author response image 1. respfig1:** Replicates used for graphs and heat maps of Figure 3. Replicate data was combined except for taf1-AA and taf1-AAR for Med15, as the second ChIP experiment failed.

Reviewer #2 was also concerned about whether data in Figure 5 was based on replicate experiments. The figure as presented in the critiqued manuscript included combined data for two experiments that were consistent with two additional replicate experiments for both Med15 and Med18 ChIP in kin28-tbp-AA yeast (four replicates in total), and data for single experiments that were consistent with replicate experiments for Med15 and Med18 ChIP in kin28-AA yeast. The reason for combining data only for kin28-tbp-AA yeast is that we obtained fewer reads for those experiments, and combining data yielded better signal to noise. For Med18 in kin28-taf1-AA and kin28-rpb3-AA strains, we showed data from single experiments that were qualitatively consistent with replicate experiments. However, the extent to which Med18 signal was affected in the kin28-taf1-AA mutant differed quantitatively in these replicate experiments. We have now performed an additional replicate of this experiment. All three replicates are qualitatively consistent (see Author response image 2): depletion of Kin28 together with Taf1 decreases signal at TFIID-dominated genes, and does not substantially shift Mediator peaks at UAS genes as observed for kin28-tbp-AA yeast (see Author response image 2). The effect of depletion of Kin28 and Taf1 on Med18 occupancy at SAGA-dominated and UAS genes (which are mostly SAGA-dominated) was variable in our three replicate experiments with regard to peak magnitude relative to depletion of Kin28 alone, although it was always a less pronounced decrease than seen for TFIID-dominated genes. We have modified Figure 5 to use the replicate experiment that shows intermediate changes in signal intensity, and make a note of this variability in the figure legend of the revised manuscript. Finally, only single replicates had been performed for Med15 ChIP using kin28-rpb3-AA and kin28-taf1-AA yeast. We have repeated these experiments and obtained consistent results in kin28-rpb3-AA yeast; however, we have seen stronger signals for Med15 in kin28-taf1-AA yeast in the presence of rapamycin than in the experiment, which was used for Figure 5 in the critiqued version (see Author response image 2). Again, we see an upstream shift upon depletion only with TBP and not with Taf1 or Rpb3, and strong decrease in signal at TFIID-dominated genes (see revised Figure 5). We have replaced the data for Med15 ChIP in kin28-taf1-AA in Figure 5 with data from one of the new replicates (the “run40” replicate shown in Author response image 2), to reflect the stronger signal we see for kin28-taf1-AAR relative to kin28-AAR in the new replicates. We have not used combined data for Figure 5 (except as noted above), as the signal to noise and number of reads has varied among replicates, and the combined data is dominated by the “cleanest” data. Information on replicate experiments provided as Supplementary file 3 has been updated to reflect these additional replicate experiments.

**Author response image 2. respfig2:** Replicate data for Figure 5.

The reviewer also states that experiments are normally done in duplicate (at least) and some correlation metrics used to insure they are alike. We have found that use of correlation coefficients between ChIP-seq datasets is not informative, as variance in regions outside of the peaks that we focus on (the large majority of the genome) obscures the similarities or differences among the regions of interest in terms of quantitative metrics. Rather, we have used metagene analysis, as in Figures 3 and 5 and the Author response image 1 and Author response image 2 to assess similarity between replicate experiments.

Another major concern with the manuscript has to do with the Taf1 anchor-away data. As acknowledged by the authors, the taf1-FRB strain appears to show a phenotype even in the absence of rapamycin. This means that the tagging affected the function of the protein. Because rapamycin did not lead to any addition effect, I do not think that the data can be interpreted as a consequence of conditional depletion. Unless the authors find a way to conditionally affect the function of Taf1 (perhaps auxin degron or tagging FRB in N-terminal would work better), I believe the Taf1 data has to be removed.

The reviewer states that “addition of rapamycin did not lead to any additional effect”, but this is not correct: although adding rapamycin did not affect the ChIP signal for Med15 or Med18 (Figure 3), it did cause lethality (Figure 3—figure supplement 1) and reduced Taf1 occupancy (Figure 3—figure supplement 3 in the revised manuscript); furthermore, anchoring away Taf1 together with Kin28 suppresses the effect of anchoring away Kin28 at TFIID-dominated genes (Figure 5). We therefore believe that the data presented support the main conclusion of experiments on the taf1-AA strain, which is that Mediator association with UAS regions does not decrease upon depletion of Taf1, and therefore that Taf1 is not required for Mediator recruitment to UAS regions. Furthermore, we have observed in three independent experiments that depletion of Taf1 together with Kin28 results in preferential decrease in Mediator occupancy at TFIID-dominated genes. However, the effect of this double depletion on Med15 occupancy at SAGA-dominated genes has been variable in our experiments, as mentioned above, and so we have tempered our conclusions regarding this specific effect.

In sum, I recommend the authors go back to the basics and make sure their experiments are reproducible. For that, they need to show that their replicates are highly correlated. At current state, I suspect that most of the effects observed are batch effects rather than biological effects. For instance, simply inspecting the red tracks in Figure 3—figure supplement 5A clearly shows that rep2 did not work nearly as well as rep1. Not seeing the labels, one could not even tell how to pair them. Also, looking at the blue tracks from the same Figures suggests that the samples cluster by batches rather than by conditions.

We have revised this Figure (now Figure 3—figure supplement 4) so that tracks are normalized across samples for each strain (e.g., all four tbp-AA strain scans are normalized) rather than only between samples within a single experiment (e.g. tbp-AA-1 and tbp-AAR-1 samples were normalized relative to one another but separately from tbp-AA-2 and tbp-AAR-2 samples). This shows more clearly the reduced signal of TBP upon addition of rapamycin.

[Editors' note: further revisions were requested prior to acceptance, as described below.]

Reviewer #2:The revised manuscript is improved, notably by the addition of datasets completing some that were done only once and by the replacement of non-consistent replicates. However, there are a few issues that are still of concern and should be addressed.First, some of the anchor-away strains behave strangely. While this is unlikely to drive erroneous conclusions, it should at least be mentioned. For instance, Kin28-TBP-AA Med15-Myc is very sick according to Figure 3—figure supplement 1. More worrisome, tagging various PIC components with FRB greatly affects Mediator ChIP in the absence of rapamycin. This is evident by looking at several figures, notably in Figure 3 (pay attention to the y-axis for all the no rapa traces (black) or look at the intensities of the no rapa heatmaps). This is also directly shown in Figure 3—figure supplement 2B. The authors use this figure supplement to claim that Taf1-AA leads to increased Mediator ChIP signal in absence of rapamycin but they fail to acknowledge that this is also the case for TBP-AA (the average signal is 2 to 3 fold up compared to non-tag). It is not clear whether these differences are batch differences (due to technical variation between experiments performed on different days) or biological differences (caused by partial loss of function of tagged proteins).

We have edited the text to note the effect of the tbp-FRB allele on Mediator occupancy and the slow growth phenotype of the kin28-tbp-AA Med15-myc strain (Results section). The elevated Med15 occupancy observed in taf1-AA yeast in the absence of rapamycin was seen in two out of two replicate experiments, while lower levels of Med15 occupancy, similar to the control lacking any FRB tag, was seen in two out of two experiments with rpb3-AA yeast and two out of three experiments in tbp-AA yeast. One experiment showed elevated Med15 occupancy in tbp-AA yeast in the absence of rapamycin; that is the experiment that led to the request for an additional replicate, which showed low occupancy in the absence of rapamycin in tbp-AA yeast that increased upon rapamycin treatment. We have also edited Figure 3—figure supplement 2 by replacing the data for the “no FRB tag” control with data from a newer replicate (from October this year). The original data had higher levels of artifactual signal over ORF regions (as we and others often see), which made it more difficult to visually compare signal strength over the UAS regions.

Second, the Med18 ChIP data displayed in Figure 3 and showing that Med18 does not change in TBP-AAR or Rpb3-AAR does not fit the author's model very well. In subsection “Mediator association at UAS regions is stabilized by loss of TBP or Pol II” the authors say "The increase in ChIP signal observed for Med15, but not for Med18, would be consistent with the most immediate contacts with UAS-bound activators occurring through tail module subunits (such as Med15), while ChIP signal for Med18 at UAS regions would depend on protein-protein cross-links, thus lowering the efficiency of ChIP for this subunit". While I agree that this explains the lower ChIP efficiency of Med18 at UAS, it does not explain why, unlike other subunits tested, Med18 signal does not increase at UAS in TBP-AAR and Rpb3-AAR. The observed Med18 ChIP signal is clearly over the UAS so if Mediator is stuck there, this Med18 signal should increase together with other subunits. The fact that it does not (assuming the data is real), suggests a more complicated model. This result should at least be described more critically.

We agree with this criticism, and have modified the text accordingly. We did not make our point clearly, which is that under some circumstances, such as depletion of Kin28 (Figure 3—figure supplement 4B, and also Jeronimo et al., 2016), Med15 can be detected at UAS regions while Med18 gives rise to virtually no ChIP signal at those sites; in the same sample, signals from both Med15 and Med18 are easily discerned at the proximal promoter. This suggests that Mediator may adopt a configuration in which Med15 signal is robust and Med18 undetectable, or nearly so; such a configuration could result in increased Med15 signal with no increase in Med18 signal. The way we presented this before was poorly constructed at best, and we thank the reviewer for noticing this.

Third, the conclusion that TBP is required for the transition while Rpb3 and Taf1 are stabilising the PIC-associated Mediator is in my opinion a hazardous extrapolation. As explained by the authors in subsection “TBP is required for transit of Mediator from UAS to promoter”, depletion of PIC components has impact on other PIC components. This makes the attribution of specific roles to specific PIC components in Mediator dynamics difficult. Hence, with these limitations in mind, it seems like an over interpretation to say "TBP being critical for transit of Mediator from UAS to promoter, while Pol II and Taf1 stabilize Mediator association at proximal promoters". A more conservative interpretation would be that the PIC is necessary to allow Mediator transiting to the promoter (as shown when PIC is severely compromised; TBP-AA) and that partially formed PICs (without trying to attribute specific roles to subunits or proteins), while allowing the transit, lead to unstable Mediator-PIC association.These last two points had been raised in previous review rounds but are still not clearly dealt with in the most recent version of the manuscript.

We believe that our statement that TBP is critical for transit of Mediator from UAS to promoter is logically accurate. It does not imply that other factors, in this case other PIC components that depend on TBP for stable association with promoters do not contribute. As we state in subsection “TBP is required for transit of Mediator from UAS to promoter”, we can infer that TBP is necessary for Mediator transit to promoter regions under conditions of Kin28 depletion, but not whether it is sufficient. In contrast, the evidence presented indicates that under conditions of Kin28 depletion, neither Taf1 nor Pol II are critical for this transit. However, the reviewer’s summary statement does provide a succinct summary of these points, and we have edited the text to borrow his/her phrasing (hopefully with permission).

Finally, it is not clear why the authors were expecting a decrease of Mediator at UAS upon depletion of TBP, Rpb3, or Taf1. Isn't an increase what the "dynamic model" proposed by the Struhl and Robert labs predicted?

We had initially anticipated that PIC depletion might reduce Mediator occupancy at both promoters and UAS regions. This could occur, for example, if Mediator contacted both UAS and promoter via DNA looping and reduced PIC occupancy destabilized Mediator occupancy at both regions. This was evidently not the right way to think about it, but it still seems to me not a totally far-fetched possibility in the absence of data. We have edited the text to clarify this point. Clearly what we anticipated was not as important in any event as what we observed.